# The immediate-early protein 1 of human herpesvirus 6B interacts with NBS1 and inhibits ATM signaling

Vanessa Collin [1,2,7], Élise Biquand [3,4,5,6,7], Vincent Tremblay[3,4,5,7], Élise G Lavoie [3,5], Andréanne Blondeau [3,5], Annie Gravel [1,2], Maxime Galloy [3,4,5], Anahita Lashgari [3,4,5], Julien Dessapt [3,4,5], Jacques Côté [3,4,5], Louis Flamand [1,2 ✉] & Amélie Fradet-Turcotte [3,4,5 ✉]

## Abstract

**Viral infection often trigger an ATM serine/threonine kinase (ATM)-dependent DNA damage response in host cells that suppresses viral replication. Viruses evolved different strategies to counteract this antiviral surveillance system. Here, we report that human herpesvirus 6B (HHV-6B) infection causes genomic instability by suppressing ATM signaling in host cells. Expression of immediate-early protein 1 (IE1) phenocopies this phenotype and blocks homology-directed double-strand break repair. Mechanistically, IE1 interacts with NBS1, and inhibits ATM signaling through two distinct domains. HHV-6B seems to efficiently inhibit ATM signaling as further depletion of either NBS1 or ATM do not significantly boost viral replication in infected cells. Interestingly, viral integration of HHV-6B into the host's telomeres is not strictly dependent on NBS1, challenging current models where integration occurs through homology-directed repair. Given that spontaneous IE1 expression has been detected in cells of subjects with inherited chromosomally-integrated form of HHV-6B (iciHHV-6B), a condition associated with several health conditions, our results raise the possibility of a link between genomic instability and the development of iciHHV-6-associated diseases.**

**Keywords** DNA Double-Strand Break Signaling; Telomere; Integration; Human Herpesvirus 6A/B; Immediate-Early Protein IE1
**Subject Categories** DNA Replication, Recombination & Repair; Microbiology, Virology & Host Pathogen Interaction; Signal Transduction

## Introduction

To infect a cell, a virus needs to successfully replicate its genetic material and produce new virions. Accordingly, cells have a sophisticated surveillance system that detects viral DNA and activates an innate antiviral response. Mounting evidence supports a role for the DNA damage response (DDR) in this process, revealing an intricate interplay between its activation and the activation of intrinsic antiviral responses (Justice and Cristea, 2022). Some viruses, such as adenovirus, target the MRE11-RAD50-NBS1 (MRN) complex for degradation to prevent the activation of ATM serine/threonine kinase (ATM) (Stracker et al, 2002; Lakdawala et al, 2008), while others, such as herpes simplex virus 1 (HSV-1) and human papillomavirus, rely on these proteins for efficient viral replication (Lilley et al, 2005; Moody and Laimins, 2009; Anacker et al, 2014). How and why viruses inhibit or hijack the ATM pathway remains a mystery (Weitzman and Fradet-Turcotte, 2018).

In mammalian cells, the MRN complex and ATM are essential to maintain genomic stability in the presence of DNA double-strand breaks (DSBs). Broken DNA ends are first detected by the MRN complex (Syed and Tainer, 2018), where its accumulation induces a signaling cascade that activates ATM and subsequent phosphorylation of the histone variant H2AX on Ser139 (producing γH2AX). Mediator of DNA damage checkpoint 1 (MDC1) interacts with γH2AX, triggering the ubiquitylation of chromatin surrounding the break by promoting the accumulation of the E3-ubiquitin ligases ring finger protein (RNF) 8 and RNF168 (Fradet-Turcotte et al, 2013; Mattiroli and Penengo, 2021). In the G1 phase of the cell cycle, the recruitment of the DNA repair factor tumor protein p53 binding protein 1 (53BP1) at ubiquitylated chromatin promotes DNA repair via non-homologous end-joining (NHEJ) (Krenning et al, 2019). In the S and G2 phases, BRCA1 DNA repair associated (BRCA1) and RB binding protein 8 endonuclease (CtIP) accumulate at the break and cooperate with exonuclease 1 (EXO1), BLM RecQ like helicase, and DNA replication helicase/nuclease 2 (BLM-DNA2) to facilitate end resection. This process results in extensive single-stranded (ss) DNA accumulation, which ultimately triggers the recruitment of recombinases that drive the homology searching required for homology-driven recombination (HDR) (Maréchal and Zou, 2013). HDR uses homologous sequences as templates to repair breaks in a faithful manner and includes processes such as homologous recombination (HR),

[1]Division of Infectious Disease and Immunity, Centre Hospitalier Universitaire (CHU) de Québec-Université Laval Research Center, Quebec City, QC G1V 4G2, Canada. [2]Department of Microbiology, Infectious Disease and Immunology, Faculty of Medicine, Université Laval, Quebec City, QC G1V 0A6, Canada. [3]Oncology Division, Centre Hospitalier Universitaire (CHU) de Québec-Université Laval Research Center, Quebec City, QC G1R 2J6, Canada. [4]Department of Molecular biology, Medical Biochemistry and Pathology, Faculty of Medicine, Université Laval, Quebec City, QC G1V 0A6, Canada. [5]Université Laval Cancer Research Center, Université Laval, Quebec City, QC G1R 3S3, Canada. [6]Present address: INSERM, Centre d'Étude des Pathologies Respiratoires (CEPR), UMR 1100, Université de Tours, Tours, France. [7]These authors contributed equally: Vanessa Collin, Élise Biquand, Vincent Tremblay. ✉E-mail: louis.flamand@crchudequebec.ulaval.ca; amelie.fradet-turcotte@crchudequebec.ulaval.ca

single-stranded annealing (SSA), and break-induced replication (BIR) (Kockler et al, 2021; Bhargava et al, 2016; Krenning et al, 2019; Kramara et al, 2018).

Human herpesvirus 6B (HHV-6B) is a betaherpesvirus that infects nearly 90% of the world's population in the first 2 years of life and is responsible for roseola infantum, a pathology defined by skin rashes, high fever, and respiratory distress (Yamanishi et al, 1988; Zerr et al, 2005; Hall et al, 2004). In this double-stranded DNA (dsDNA) virus subfamily, HHV-6B shares 90% homology with HHV-6A, another lymphotropic virus. Although both viruses infect CD4[+] T lymphocytes, they have epidemiological, biological, and immunological differences (Collin and Flamand, 2017). Like other herpesviruses, HHV-6A and HHV-6B (HHV-6A/B) establish lifelong latency in infected hosts and can occasionally reactivate (Pantry and Medveczky, 2017). However, whereas most herpesviruses achieve latency by circularizing and silencing their genome, HHV-6A/B integrate their genomes into the host's chromosomal terminal repeats (telomeres) (Arbuckle et al, 2013, 2010). The linear dsDNA genomes of HHV-6A/B are both flanked by an array of direct repeats containing 15–180 reiterations of perfect telomeric repeats (pTMRs) identical to the human telomeric sequence (5′-TTAGGG-3′), enabling viral integration (Wallaschek et al, 2016b). The integration of the HHV-6A/B genomes depends on the integrity of these pTMRs (Wallaschek et al, 2016b), supporting a model in which viral integration is mediated through HDR, including the SSA or BIR DNA repair mechanisms (Aimola et al, 2020). When HHV-6A/B integration occurs in a gamete before fertilization, the newborn carries a copy of HHV-6A/B in every cell of its body and can be transmitted to its offspring. This condition, called inherited chromosomally-integrated (ici)HHV-6A/B, affects ~1% of the world's population, representing almost 80 million people (Tanaka-Taya et al, 2004; Daibata et al, 1999). It is more prevalent in those suffering from health issues such as high spontaneous abortion rates (Minocherhomji et al, 2015; Miura et al, 2021), pre-eclampsia (Gaccioli et al, 2020), and angina pectoris (Gravel et al, 2015) compared to healthy subjects (reviewed in Pellett, 2012; Flamand, 2018). However, the reason why iciHHV-6A/B contributes to these clinical syndromes has not been elucidated in any detail.

HHV-6B, which is better characterized than HHV-6A, sequentially expresses more than 97 genes/proteins during its lytic cycle (Gravel et al, 2021). Following viral entry, immediate-early (IE) proteins are the first proteins to be expressed (Øster and Hoëllsberg 2002). They are essential to regulate viral gene expression and establish a favorable environment for viral replication. In the context of HHV-6B, IE protein 1 (IE1) is the first protein expressed during cell infection (Schiewe et al, 1994). It inhibits the innate antiviral response in part by sequestering signal transducer and activator of transcription 2 (STAT2) in the nucleus, thereby compromising type I interferon production and signaling (Jaworska et al, 2007, 2010). In infected cells, IE1 is exclusively localized within PML bodies (Bernardi and Pandolfi, 2007), which were recently implicated in HDR-mediated DNA repair through an undefined mechanism (Yeung et al, 2012; Attwood et al, 2020; Dellaire et al, 2006; Vancurova et al, 2019). Interestingly, PML depletion reduces HHV-6B integration (Collin et al, 2020), suggesting that IE1-containing PML bodies also participate in viral integration.

In this study, we report that HHV-6B infection—and more specifically, IE1 expression—leads to genomic instability in cells. Further investigations revealed that IE1 specifically prevents phosphorylation of the histone variant H2AX and subsequent HDR repair. Structure-function analyses reveal that the N-terminal domain of IE1 interacts with the BRCA1 C-terminal domain 2 of nibrin (NBN, also known as NBS1) and this interaction is independent of the induction of DNA damage. In contrast, the C-terminal domain of IE1 drives the inhibition of ATM both in cells and in vitro. Consistent with the infection being sufficient to efficiently inhibit ATM signaling, we show that HHV-6B replication in infected cells is not further increased upon depletion of NBS1 or ATM. Although current models propose that viral integration occurs through HDR DNA repair, we show that viral integration is not affected in NBS1-depleted cells that elongate their telomeres in a human telomerase reverse transcriptase (hTERT)-dependent manner. Thus, our findings reveal that viral integration relies on biological pathways that safeguard telomere extension in infected cells and not on specific DNA repair pathways.

# Results

## HHV-6B infection and IE1 expression induce genomic instability

Our first indication that HHV-6B infection induces genomic instability in host cells was the observation that HHV-6B infection rapidly induced micronuclei (MNi) formation in MOLT-3 cells (a lymphoblast T cell line; Fig. 1A and Appendix Fig. S1A). In these experiments, cells infected with the HHV-6B strain Z29 accumulated 6.6-fold more MNi than non-infected cells (Mock) 24 h post-infection (Fig. 1A). Consistent with the rapid accumulation of genomic instability in infected cells, we observed a similar phenotype in duplicate clones of stable U2OS cell lines containing a doxycycline (Dox)-inducible expression cassette for IE1 (C10 and C102; Fig. 1B and Appendix Fig. S1B,C). These clones had 4.8-fold more MNi than parental U2OS cells 48 h post-IE1 induction (Fig. 1B), suggesting that the instability observed in HHV-6B infected cells is at least partially caused by IE1. Note that both U2OS clones (C10 and C102) exhibit similar levels of MNi than the parental cell line without IE1 induction (Appendix Fig. S1C). MNi arise from unresolved genomic instabilities such as DSBs (i), lagging chromosomes (ii), and ruptured anaphase bridges (Abs) (iii) (Fig. 1C) (Cassel et al, 2014). They are compartmentally separated from the primary nucleus and surrounded by a nuclear envelope, as shown by the presence of lamin B in these perinuclear structures (Appendix Fig. S1D,E; Hatch et al, 2013). To determine how IE1 triggers MNi formation, we analyzed the accumulation of different markers in IE1-induced MNi, such as centromeres and telomeric DNA (to detect lagging chromosomes and Abs caused by telomere fusion, respectively). Interestingly, a much lower proportion of the IE1-induced MNi contained centromeres compared with those in parental U2OS cells (~10-fold, Fig. 1D and Appendix Fig. S1F), suggesting that they are not induced by chromosome segregation defects. Although IE1 partially colocalizes with telomeres in host cells (Collin et al, 2020), fluorescence in situ hybridization (FISH) revealed that IE1 does not specifically promote instability at telomeres, as IE1-induced MNi accumulated similar levels of telomeric DNA as those in parental U2OS cells (Fig. 1E and Appendix Fig. S1G). Metaphase spread assays revealed that IE1-expressing U2OS cell lines exhibited higher frequencies of

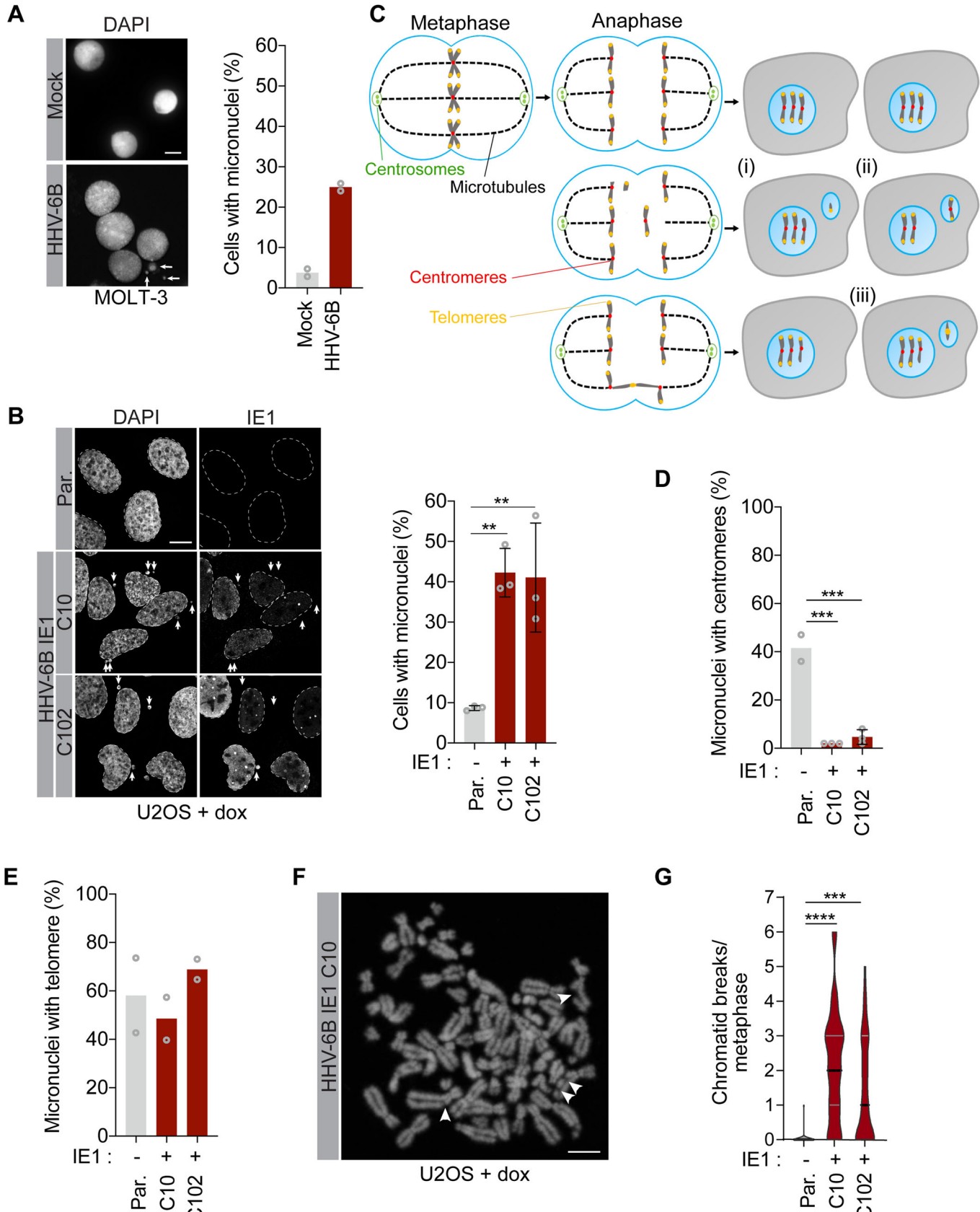

**Figure 1. HHV-6B infection and IE1 expression lead to micronuclei formation.**

(A) Left panel: Representative images of Mock and HHV-6B-infected MOLT-3 cells (MOI of 1), fixed 24 h post-infection and counterstained with DAPI. Micronuclei are indicated by white arrows. The quantification of the micronuclei (right panel) shows the mean ($n = 2$ biological replicates, >100 micronuclei/condition). (B) Left panel: representative images of U2OS cell lines and clones stably expressing Dox-inducible HHV-6B IE1 (C10 and C102). IE1 expression was induced with 1 µg/mL Dox for 48 h prior to IE1 immunofluorescence. Micronuclei are indicated by white arrows. The parental cell line (Par.) was used as a negative control. Quantification is provided in the right panel ($n = 3$ biological replicates). (C) Schematic of micronuclei formation via DNA double-strand breaks (DSBs) (i), lagging chromosomes (ii), and anaphase bridges (ABs) (iii). (D, E) Quantification of micronuclei containing centromeres (D, $n \geq 2$, biological replicates) and telomeres (E, $n = 2$ biological replicates, >100 micronuclei/condition). Cells were treated as in (B) and centromeres and telomeres were detected by immunofluorescence and FISH, respectively. (F) A representative metaphase spread from an IE1-expressing cell. Cells were treated with 1 µg/mL Dox for 48 h, then metaphase spreads were prepared, fixed, and counterstained with DAPI. Chromosomal aberrations are indicated by white arrows. (G) Quantification of chromosomal aberrations per metaphase ($n = 31$ from 3 biological replicates). Data information: In (B, D, E, G), data are presented as mean ± SD. **$p \leq 0.01$, ***$p \leq 0.001$, ****$p \leq 0.0001$ (One-way ANOVA with Dunnett's multiple comparison test). Scale bars = 5 µm. Source data are available online for this figure.

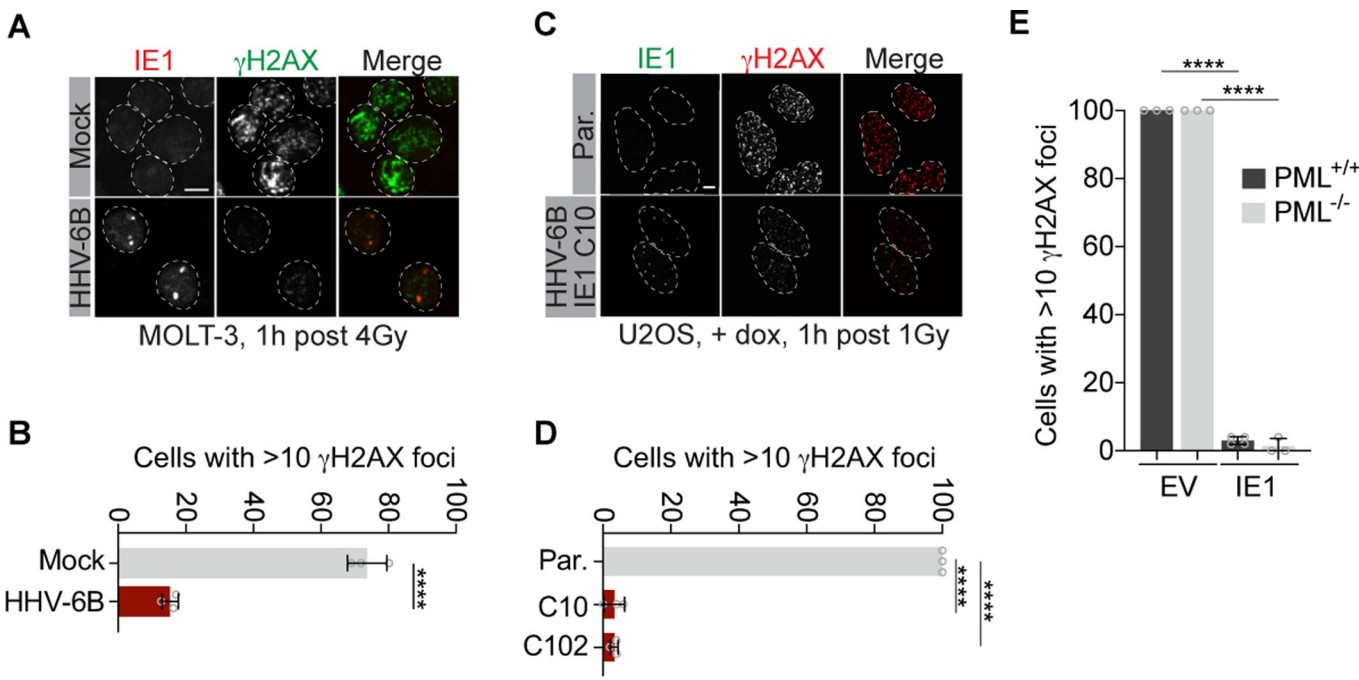

**Figure 2. H2AX phosphorylation (γ-H2AX) is inhibited in HHV-6B-infected and IE1-expressing cells.**

(A) Representative γH2AX immunostaining in HHV-6B-infected MOLT-3 cells irradiated with 4 Gy and immunostained for IE1 and γH2AX 1 h later. Mock-infected cells were used as a negative control. (B) Quantification of irradiated MOLT-3-infected and Mock cells with >10 γH2AX foci ($n = 3$ biological replicates). (C) Representative γH2AX immunostaining in irradiated U2OS parental (Par.) and IE1-expressing cells (clone C10). IE1 expression was induced as in Fig. 1B. Cells were irradiated with 1 Gy and immunostained for IE1 and γH2AX 1 h later. (D) Quantification of irradiated U2OS Par. and IE1-expressing cells with >10 γH2AX foci ($n = 3$ biological replicates). (E) Quantification of cells with >10 γH2AX foci in irradiated (1 Gy) U2OS PML$^{+/+}$ and PML$^{-/-}$ cells that transiently express untagged IE1. An empty vector (EV) was used as a negative control ($n = 3$ biological replicates). Data information: In (B, D, E), data are presented as mean ± SD. ****$p \leq 0.0001$ (B: unpaired $t$ test, D, E: One-way ANOVA with Dunnett's multiple comparison test). Scale bars = 5 µm. Source data are available online for this figure.

DNA breaks than parental cells (Fig. 1F,G), consistent with MNi being induced by DSB accumulation. Interestingly, IE1 was detected in only 5–10% of the MNi (Appendix Fig. S1H), suggesting that they are not arising from DSBs induced by the physical binding of IE1 at any defined DNA locus.

## HHV-6B IE1 impairs DSB signaling and homology-directed DNA repair

DSB accumulation results from either an increase in DNA breaks or defective DNA DSB signaling and repair. To determine how HHV-6B infection promotes genomic instability, we first investigated whether infected cells accumulated Ser139-phosphorylated H2AX (i.e., the DSB marker γH2AX). Surprisingly, while γH2AX foci accumulated in >75% of non-infected MOLT-3 cells exposed to irradiation (IR), this number was dramatically reduced in infected cells (Fig. 2A,B and Appendix Fig. S2A). U2OS clones stably expressing IE1 reproduced this phenotype (Fig. 2C,D and Appendix Fig. S2B), indicating that IE1 impairs DSB signaling. This inhibition is independent of IE1 accumulation within PML bodies, as no γH2AX foci were detected in PML-deficient U2OS cells that transiently express IE1 (PML$^{-/-}$, Fig. 2E and Appendix Fig. S2C–E).

DSB signaling is essential to activate DNA repair pathways. Therefore, we tested whether DNA repair is inhibited in HHV-6B IE1-expressing cells. As mammalian cells use several pathways to

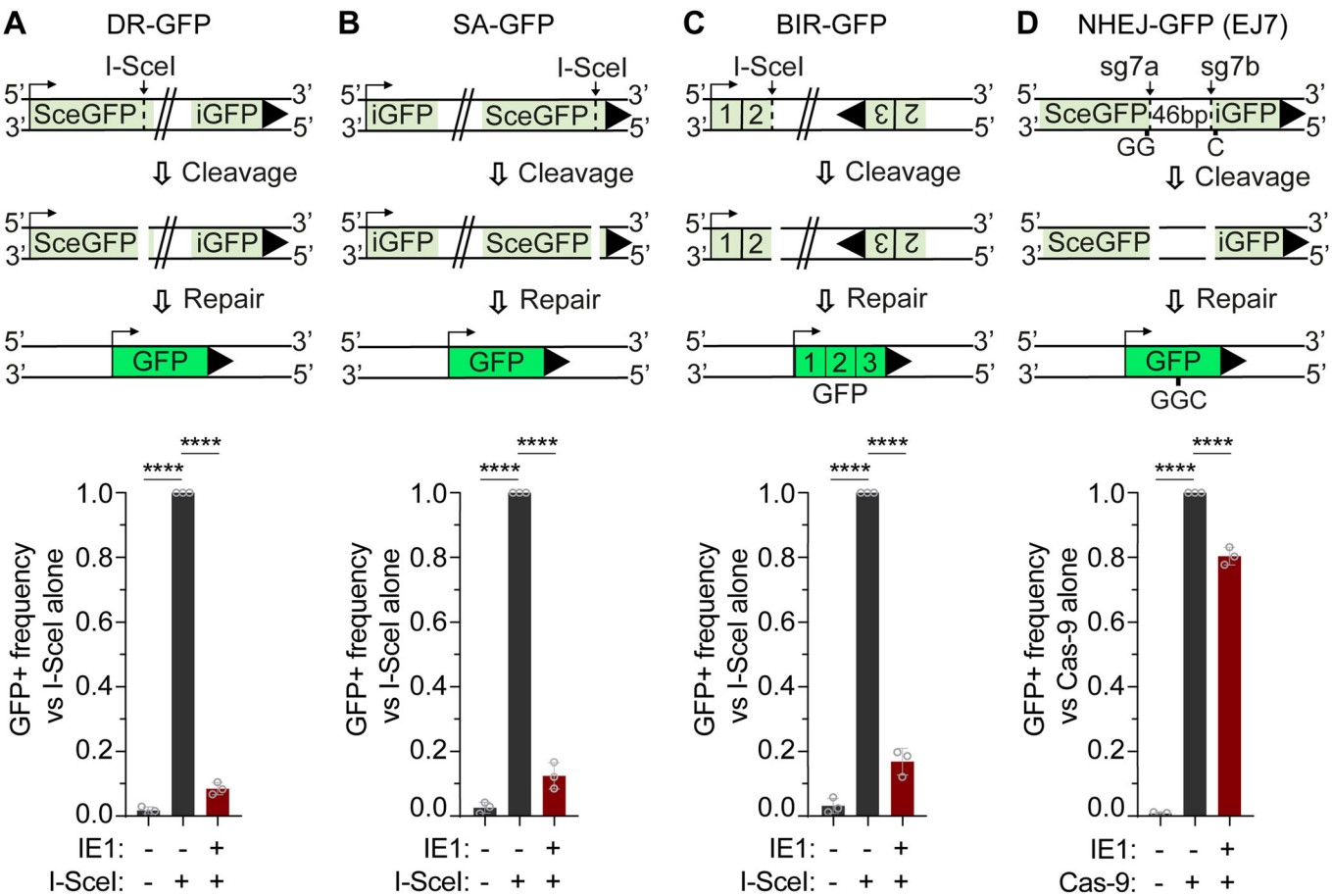

**Figure 3. HHV-6B IE1 inhibits HDR-mediated repair.**

(A–D) DNA repair reporter assays for (A) homologous recombination (DR-GFP), (B) single-strand annealing (SA-GFP), (C) break-induced replication (BIR-GFP), and (D) non-homologous end-joining (NHEJ-GFP (EJ7)). For each assay, a schematic is presented in the top panel and the flow cytometry-based quantification of GFP+ cells is presented in the bottom panel. In each replicate, GFP+ cells were normalized to the GFP+ cells in the positive control (I-SceI+ or Cas9+, set to 1.0) ($n = 3$ biological replicates). Data information: In (A–D), data are presented as mean ± SD. ****$p \le 0.0001$ (One-way ANOVA with Dunnett's multiple comparison test). Source data are available online for this figure.

repair DSBs (Hustedt and Durocher, 2016), we performed a panel of DNA repair reporter assays in cells with stable or transient IE1 expression. These reporter assays all rely on the detection of a fluorescent protein that is expressed only if a site-specific DSB (induced by either I-SceI or Cas-9) is adequately repaired (Fig. 3A–D and Appendix Fig. S3A, top panels) (van de Kooij and van Attikum, 2022; Pinder et al, 2015). In the DR-GFP (direct repeats) and CRISPR-LMNA HDR assays, DSBs repaired by HR either reconstitute a defective green fluorescent protein (GFP)-reporter transgene integrated into the genome (DR-GFP) or introduce a cassette expressing mRuby in frame with endogenous lamin A (CRISPR-LMNA HDR), respectively (Stark et al, 2004; Boonen et al, 2019). In SA-GFP (single-strand annealing), proper annealing of a small homologous stretch reconstitutes a truncated GFP (Stark et al, 2004). In BIR-GFP (break-induced replication), replication-mediated repair following homology searching places a GFP coding sequence in the correct orientation (Costantino et al, 2014; Kramara et al, 2018). Finally, in NHEJ-GFP, the ends of two DSBs need to be correctly ligated to recreate a full-length (FL) GFP-expressing cassette (Jacquet et al, 2016; Bhargava et al, 2018).

In assays using reporter transgenes integrated into the genome (DR-GFP, SA-GFP, BIR-GFP, and NHEJ-GFP) (Fig. 3A–D and Appendix Fig. S3B,C), a condition without endonuclease I-SceI or Cas-9 was used as a negative control and the percentage of fluorescent cells obtained with I-SceI was set to 1. In each condition, a small amount of a near-infrared fluorescent protein (iRFP)-expressing vector was transfected with I-SceI, IE1, or an empty vector (EV) to ensure that DNA repair was measured in transfected cells only, and DNA repair was assessed 48 or 72 h post-transfection. Finally, as the clonal BIR-GFP U2OS cell line was generated in this study using a previously described BIR-GFP reporter plasmid (Costantino et al, 2014), we used short interfering (si)RNAs against RAD51 and RAD52 as additional controls (Appendix Fig. S3D–F; Kockler et al, 2021). As expected, the BIR-GFP signal was specifically inhibited in cells depleted of RAD51 (Costantino et al, 2014).

Interestingly, these analyses revealed that both transient and stable IE1 expression drastically reduced all types of homology-directed DNA repair (Fig. 3A–C, lower panels, and Appendix Fig. S3A–C). In contrast, IE1 only slightly modulated DNA repair

in reporter assays assessing NHEJ (Fig. 3D). As the choice between HDR and NHEJ is driven by the cell cycle (Hustedt and Durocher, 2016), we confirmed that cell cycle progression was not affected in cells expressing IE1 (Appendix Fig. S3G). Altogether, these results show that homology-based DNA repair is specifically inhibited in cells expressing HHV-6B IE1.

## HHV-6B IE1 interacts with NBS1 and inhibits its ability to promote ATM activation

At DSBs, homology-based DNA repair is initiated when lesions are detected by the MRN complex, which leads to ATM auto-activation (Fig. 4A) through a still poorly understood mechanism (Warren and Pavletich, 2022; Syed and Tainer, 2018; Schiller et al, 2012). In adenovirus-infected cells, ATM activation is inhibited through the degradation of the MRN complex, which is mediated by the protein E4 (Stracker et al, 2002). In contrast, MRN complex components were stable at steady state upon IE1 induction in U2OS clones (Fig. 4B). Interestingly, immunofluorescence analyses revealed that IE1 colocalizes with NBS1 in ~75% of IE1-expressing cells (Fig. 4C–E and Appendix Fig. S4A,B). This colocalization was also detected in irradiated cells and *PML* knockout cells (Appendix Fig. S4C–E), suggesting that the interaction between IE1 and NBS1 is constitutive and independent of PML bodies. We also observed constitutive colocalization between MRE11 and IE1 (Fig. 4D,E and Appendix Fig. S4A,B,F). However, the colocalization of transiently expressed IE1 with MRE11 was greatly reduced upon NBS1 depletion (Fig. 4F and Appendix Fig. S4G,H), supporting a model where IE1 colocalizes with the MRN complex by interacting with NBS1. Importantly, we confirmed that colocalization between IE1 and NBS1 is also observed in HHV-6B-infected MOLT-3 cells (Fig. 4G,H).

To further characterize the interplay between IE1 and the MRN complex, we took advantage of a cell-based assay that quantifies the ability of a mCherry-LacRnls fusion protein to specifically induce the recruitment of a "prey" to a *LacO* array integrated at a single locus in U2OS 2-6-5 cells ((i) No DSBs, Fig. 5A; Tang et al, 2013; Sitz et al, 2019). This system can also be used to study signaling at DSBs by recruiting the nonspecific nuclease domain of the restriction endonuclease FokI fused to the mCherry-LacRnls (ER-mCherry-LacR-FokI-DD) to the *LacO* array ((ii) Localized DSBs, Fig. 5A). Although ER-mCherry-LacR-FokI-DD is constitutively expressed in U2OS 2-6-5 cells, the protein is cytoplasmic, and a C-terminal destabilization domain (DD) ensures its continual degradation (Tang et al, 2013). DSBs can be rapidly induced by adding 4-hydroxytamoxifen (4-OHT) and Shield-1 to the culture medium. 4-OHT induces nuclear relocalization of ER-mCherry-LacR-FokI-DD via its modified estrogen receptor (ER) domain and Shield-1 stabilizes it by inactivating the DD. When the mCherry-LacRnls-IE1 fusion protein was transiently expressed in U2OS 2-6-5 cells, only NBS1 was efficiently recruited to the *LacO* (Fig. 5B,C and Appendix Fig. S5A–E). No DDR signaling proteins were recruited to the *LacO* by the negative control, mCherry-LacRnls. As a positive control, we added 4-OHT and Shield-1 to the medium for 6 h, and readily detected ATM, phosphorylated ATM (pATM) (Ser1981), γH2AX, RAD50, NBS1, and MRE11. No pATM or γH2AX signals were detected at the array upon recruitment of mCherry-LacRnls-IE1, consistent with a constitutive interaction between IE1 and NBS1 that is independent of DSB signaling.

Intriguingly, while an mCherry-LacRnls-NBS1 fusion protein is sufficient to induce ATM recruitment and activation, as well as subsequent H2AX phosphorylation at the *LacO* array in NIH-3T3 cells (Soutoglou and Misteli, 2008), NBS1 recruitment by mCherry-LacRnls-IE1 did not trigger ATM activation (Fig. 5B and Appendix Fig. S5B). To further validate that IE1 inhibits the ability of NBS1 to activate ATM at the *LacO*, we transiently transfected a mCherry-LacRnls-NBS1 fusion protein and an untagged or FLAG-tagged IE1 vector in U2OS 2-6-5 cells. As expected, a full length (FL) NBS1 construct (aa 1–754) specifically promoted ATM and H2AX phosphorylation at the *LacO* in approximately 75% and 50% of cells, respectively (Fig. 5D–F and Appendix Fig. S5F). This function depends on its ability to bind ATM, as an NBS1 ATM binding deficient construct (ΔA; aa 1–733) produced similar γH2AX levels as the negative control (Fig. 5D and Appendix Fig. S5F). Interestingly, NBS1-dependent accumulation of γH2AX and pATM was strongly inhibited in cells expressing untagged or FLAG-tagged IE1 (Fig. 5D,F and Appendix Fig. S5F–H). These findings suggest that the interaction between IE1 and NBS1 at the array directly prevents ATM activation and subsequent H2AX phosphorylation. In support of this model, we found that IE1 accumulates at the *LacO* array upon DSB induction in an NBS1-dependent manner (Fig. 5G,H), an observation that can only be made in this system, as no marker of DSB signaling can be used to detect IE1 accumulation at endogenous DSBs. In these conditions, IE1 colocalizes with 60% of mCherry-LacR-FokI foci and this amount is reduced to 30% in cells treated with a siRNA against NBS1.

## The N-terminus of IE1 interacts with the BRCT2 domain of NBS1

The functional domains of IE1 are not well characterized aside from its STAT2 binding domain (aa 270–540; Fig. 6A; Jaworska et al, 2010). Guided by its secondary structure, we designed a series of IE1 fragments that we fused to mCherry-LacRnls to assess their ability to recruit endogenous NBS1 to the *LacO* array (Appendix Fig. S6A). The fusion encoding aa 1–540 was the smallest fragment capable of recruiting NBS1 at the *LacO* array as efficiently as FL IE1 (~81% of mCherry-LacRnls-IE1 1–540 colocalized with NBS1; Fig. 6B,C and Appendix Fig. S6B). All attempts to generate smaller fragments of this N-terminal domain of IE1 resulted in unstable proteins in our hands. Interestingly, the 1–540 fragment was also the smallest to efficiently accumulate in PML bodies (Appendix Fig. S6C,D) suggesting that both functions are related. Consistently, we observed a reduced accumulation of IE1 at PML bodies in cells treated with siNBS1 (Appendix Fig. S6E,F). Previously, we show that IE1 contains a SUMOylation interacting motif (SIM) and a SUMOylation site on Lysine 802 (K802) (Collin et al, 2020; Fig. 6A) that are located outside the domain of interaction with NBS1. As expected, mutation of either the SIM, the SUMOylation site (K802R) or both have no impact on the interaction between IE1 and NBS1 (Appendix Fig. S6G–I). NBS1 encodes a 95-kDa protein with multiple domains, which are required for its recruitment to DSBs and its interactions with ATM and ATR (Bian et al, 2019). Briefly, NBS1 contains a forkhead-associated (FHA) domain and two breast cancer C-terminal domains (BRCTs) that are required for optimal phospho-dependent NBS1 accumulation at DNA breaks. The C-terminus contains a domain that promotes its interactions with MRE11 (MRE11-binding motif, MRE11-BM) and

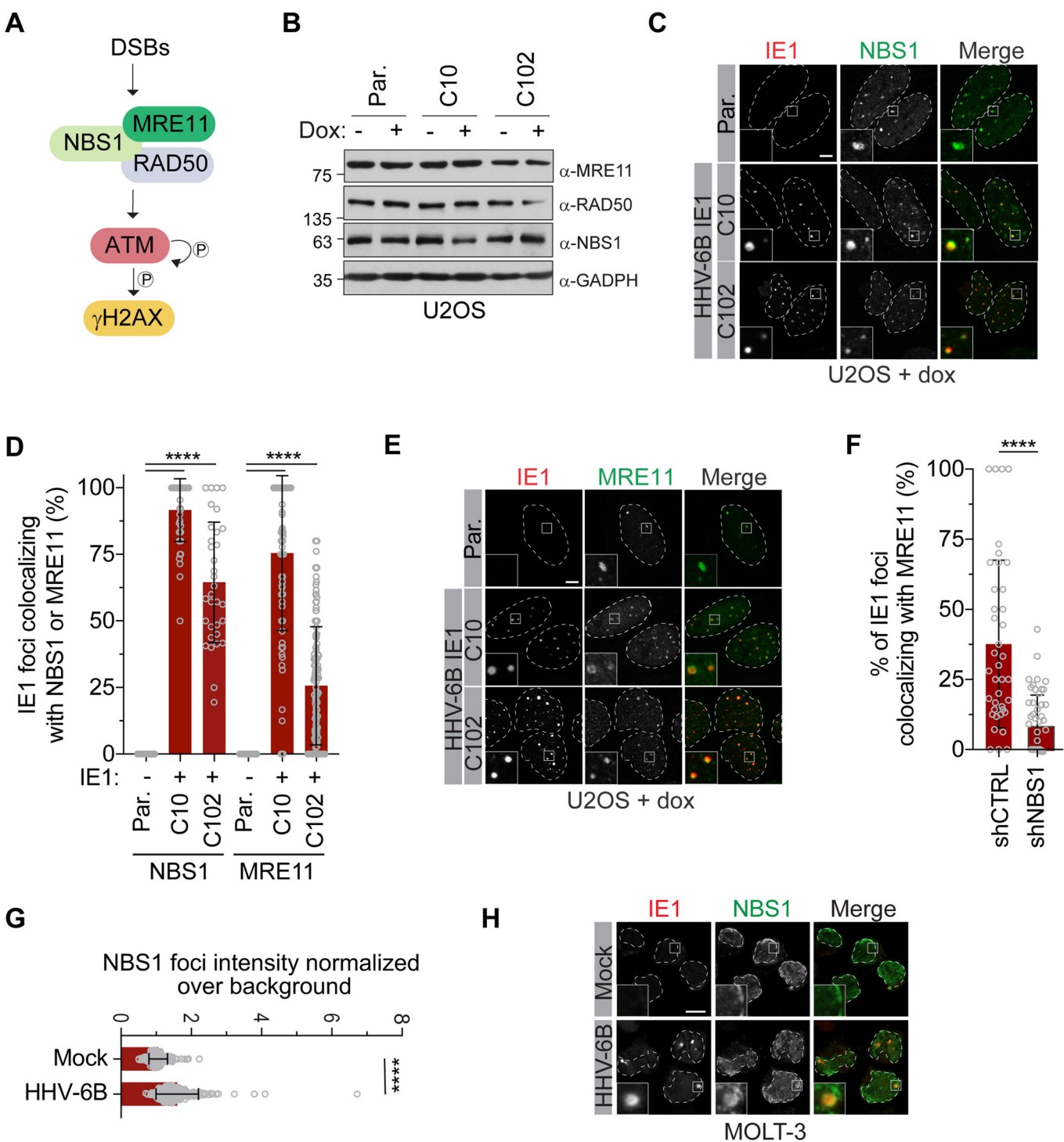

**Figure 4. HHV-6B IE1 colocalizes with NBS1.**

(A) Signaling events triggered by DNA DSBs. (B) Whole cell extracts (WCEs) from U2OS cells (Par.) and IE1-expressing U2OS stable cell lines treated with or without 1 µg/mL Dox were immunoblotted for RAD50, NBS1, and MRE11. GAPDH was used as a loading control. (C, E) Representative images of the colocalization between IE1 and NBS1 (C) and MRE11 (E). IE1-expressing cells were treated as described in Fig. 1B and immunostained for IE1, NBS1, or MRE11. As a positive control, irradiated U2OS cells (+IR) were fixed 15 min post-irradiation (1 Gy) and immunostained as indicated ("Methods," Appendix Fig. S4A). The parental cell line (Par.) was used as a negative control. (D) Percentages of IE1 foci that colocalized with NBS1 (C) and MRE11 (E) (n = 3 biological replicates). (F) Percentage of IE1 foci that colocalized with MRE11 in stable U2OS control cells (shCTRL) or those depleted of NBS1 (shNBS1) (n = 2 biological replicates, at least 40 nuclei/condition). (G, H) Quantification (G) and representative image (H) of NBS1 foci intensity in MOLT-3 mock or HHV-6B-infected cells (n = 2 biological replicates >100 foci/conditions). Data information: In (D, F, G), data are presented as mean ± SD. ****p ≤ 0.0001 (D: One-way ANOVA with Dunnett's multiple comparison test, F, G: unpaired t test). Scale bars = 5 µm. Source data are available online for this figure.

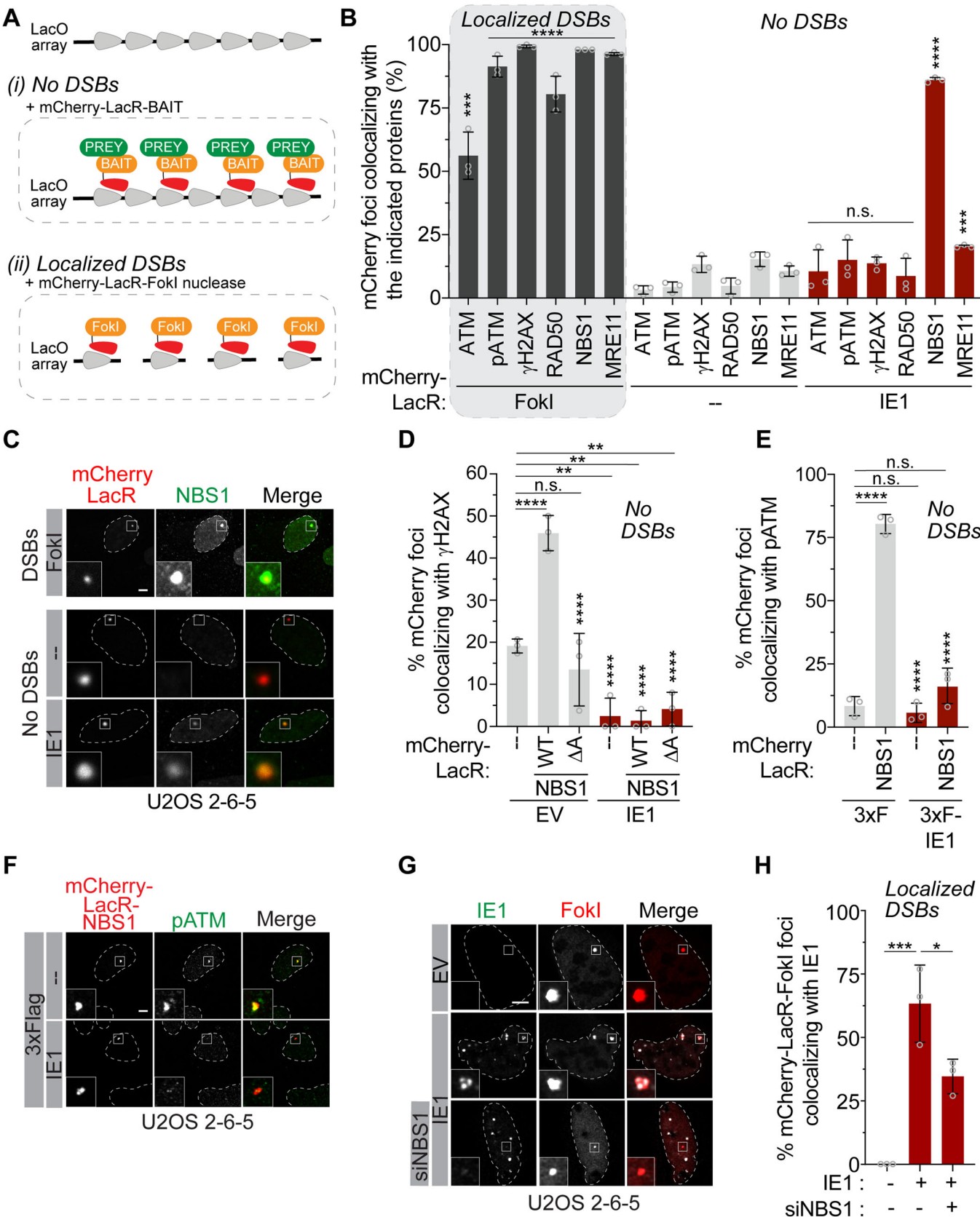

**Figure 5. HHV-6B IE1 interacts with NBS1, inhibits ATM activation, and is recruited to DSBs.**

(A) The integrated *LacO* array-based assay used to study protein colocalization at a specific locus without DSBs (i) or DNA repair protein recruitment at localized DSBs (ii). *LacO* array repeats, mCherry-LacRnls-fusion proteins, and preys are shown in grey, red/orange, and green, respectively. (B) Quantification of the indicated DSB-signaling proteins at localized DSBs induced by ER-mCherry-LacRnls-FokI-DD (FokI) or at either a mCherry-LacRnls (–, negative control) or mCherry-LacRnls HHV-6B IE1 protein foci in U2OS 2-6-5 cells. Transfected cells were treated or not with 4-OHT and Shield-1 for 6 h (FOK1 induction) then immunostained for ATM, pATM (S1981), γH2AX, RAD50, NBS1, and MRE11 (Representative images, Appendix Fig. S5A–E) (n = 3 biological replicates). (C) Representative images of NBS1 recruitment at DSBs (top panel, FOK1) and its colocalization with IE1 in the absence of DSBs (bottom panel). (D, E) U2OS 2-6-5 cells were treated as described in (B), immunostained for γH2AX (D) or pATM (E) (Representative images, Appendix Fig. S5F,G). In both experiments, cells were transfected with vectors expressing either untagged or 3×FLAG-IE1 (n = 3 biological replicates). (F) Representative images of pATM inhibition at the mCherry-LacRnls-NBS1 locus. Cells were treated as described in (B). Representative images of the negative controls are presented in Appendix Fig. S5G,H. (G) Representative images of NBS1-dependent IE1 recruitment to DSBs. Cells were treated with a siCTRL or siNBS1 prior to their transfection with an untagged IE1 or an empty vector. ER-mCherry-LacR-FokI-DD was induced as described in (B). Cells were processed for IE1 immunofluorescence. (H) Quantification of the mCherry-LacR FokI foci colocalizing with IE1 (n = 3 biological replicates). Data information: In (B, D, E, H), data are presented as mean ± SD. n.s. not significant, $*p ≤ 0.05$, $**p ≤ 0.01$, $***p ≤ 0.001$, $****p ≤ 0.0001$ (one-way ANOVA with Dunnett's multiple comparison test). In (B), statistical significance was analyzed against respective control proteins in the mCherry-LacRnls condition (–, light grey). Significance markers reported on top of the bars (vertical) indicates data compared to mCherry-LacRnls-NBS1 WT (D) and mCherry-LacRnls-NBS1 (E), respectively. Scale bars = 5 µm. Source data are available online for this figure.

ATM (ATM-binding motif, ATM-BM; Fig. 6D). Interestingly, NBS1 also contains an intrinsically disordered domain (IDD) that drives a species-specific interaction with the HSV-1 IE protein ICP0 (Lou et al, 2016). To determine if this domain also promotes the interaction between NBS1 and IE1, we used the same approach used to map the IE1-NBS1 interaction (Fig. 6A–C). Different fragments of NBS1 were fused with the mCherry-LacRnls protein and co-expressed with an untagged version of IE1 (Fig. 6D–F and Appendix Fig. S6J–O). The mCherry-LacRnls-NBS1 construct lacking the BRCT2 domain (ΔB2, Δaa 201–327) was unable to recruit IE1 to the array, while the construct containing only this domain was sufficient for the interaction (Fig. 6D–F and Appendix Fig. S6M). Consistently, immunoprecipitation of FLAG-tagged IE1 from U2OS cell lysates revealed an interaction with FL mCherry-LacRnls-NBS1 but not the ΔB2 fusion (Fig S6P). Using the LacR system, we noted that the mCherry-LacRnls-NBS1 fusion lacking the linker region of NBS1 (ΔL, Δ328–638) significantly reduced the interaction between NBS1 and IE1 (Appendix Fig. S6J,K,M,O). In contrast with the BRCT2 domain, the linker alone was unable to recruit IE1 to the *LacO* array (Appendix Fig. S6J,K,M,O). Altogether, our results support a model where the N-terminus of IE1 interacts with the BRCT2 domain of NBS1. We named this domain NBS1-binding domain (NBS1-BD) (Fig. 6A).

## The C-terminus of IE1 is sufficient to inhibit the phosphorylation of H2AX by ATM

To determine if the IE1-NBS1 interaction is sufficient to prevent ATM activation, we transiently transfected expression vectors containing the different fragments of IE1 into U2OS cells (without the *LacO* array) and quantified their ability to inhibit H2AX phosphorylation in irradiated cells. Surprisingly, the IE1 N-terminus alone was unable to prevent the accumulation of γH2AX foci (Fig. 7A,B and Appendix Fig. S7A). Instead, we found that this function depends on a fragment of 268 amino acids in the IE1 C-terminus. The 810–1078 fragment inhibited H2AX phosphorylation as efficiently as the FL protein (~75% of cells transfected with mCherry-LacRnls-IE1 810–1078 had <10 γH2AX foci 1 h post-irradiation with 1 Gy; Fig. 7B,C and Appendix Fig. S7A). Consistent with this finding, a construct of IE1 lacking amino acids 810–1078 exhibits a much weaker ability to inhibit the phosphorylation of H2AX induced by the accumulation of mCherry-LacR NBS1 at the *LacO* array (~25% inhibition instead of >80%) (Fig. 7D and Appendix Fig. S7B,C). In the context of the FL protein,

mutation of the SIM and K802 had no impact on the ability of IE1 to inhibit γH2AX (Appendix Fig. S7D,E). Together, these results show that IE1 interacts with NBS1 and blocks ATM activation using two distinct motifs: an N-terminal NBS1-binding domain and a C-terminal domain that independently inhibits the ability of NBS1 to trigger DSB signaling (Fig. 7A), which we have named ATM-inhibitory domain (ATMiD). These functions are independent of IE1 SUMOylation.

In the LacR system, IE1 did not interact with the domain of NBS1 that interacts with ATM (ATM-BM, aa 733–754) (Fig. 6D–F and Appendix Fig. S6J–O) or with ATM itself (Fig. 5B). To gain insight on the molecular mechanism by which ATMiD inhibits ATM signaling, we tested whether it can inhibit the phosphorylation of H2AX nucleosomes in a purified native ATM phosphorylation assay. In this assay, native ATM tagged at the N-terminus with 3xFLAG-2xStrep and purified by tandem affinity purification from clonal K562 cells (Dalvai et al, 2015) efficiently phosphorylates H2AX nucleosomes reconstituted with recombinant histones upon addition of ATP (Galloy et al, 2021; Fig. 7E and Appendix Fig. S7F–H). Interestingly, the addition of a 5-fold molar ratio of recombinant GST-HIS-IE1 ATMiD/H2AX in the reaction is sufficient to prevent most of H2AX phosphorylation (Fig. 7F and Appendix Fig. S7F). As the purification protocol of ATM yields low levels of ATM autophosphorylation (Fig. 7E), it is impossible to conclude whether IE1 ATMiD also inhibits ATM autophosphorylation in this assay. Unlike IE1 from human cytomegalovirus (HHV-5) (Fang et al, 2016) and the latency-associated antigen (LANA) of Kaposi's sarcoma-associated herpes virus (KSHV) (Barbera et al, 2006), the C-terminal domain of HHV-6B IE1 does not strongly interact with the nucleosome (Appendix Fig. S7I). Thus, these findings suggest that HHV-6B IE1 prevents H2AX phosphorylation through a direct inhibition of ATM.

## HHV-6B integration relies on a pathway that safeguards telomere elongation

Depending on the virus, the MRN complex is either required for viral replication or it inhibits it (Weitzman and Fradet-Turcotte, 2018). As HHV-6B IE1 interacts with NBS1 and blocks ATM activation, the complex is likely detrimental for its replication. HHV-6B infection has different outcomes depending on the nature of the infected cells (Fig. 8A). In permissive cells (e.g., MOLT-3), viral protein expression promotes replication (the lytic state). In contrast, in semi-permissive cells, integration of the viral genome into the host's telomeres is

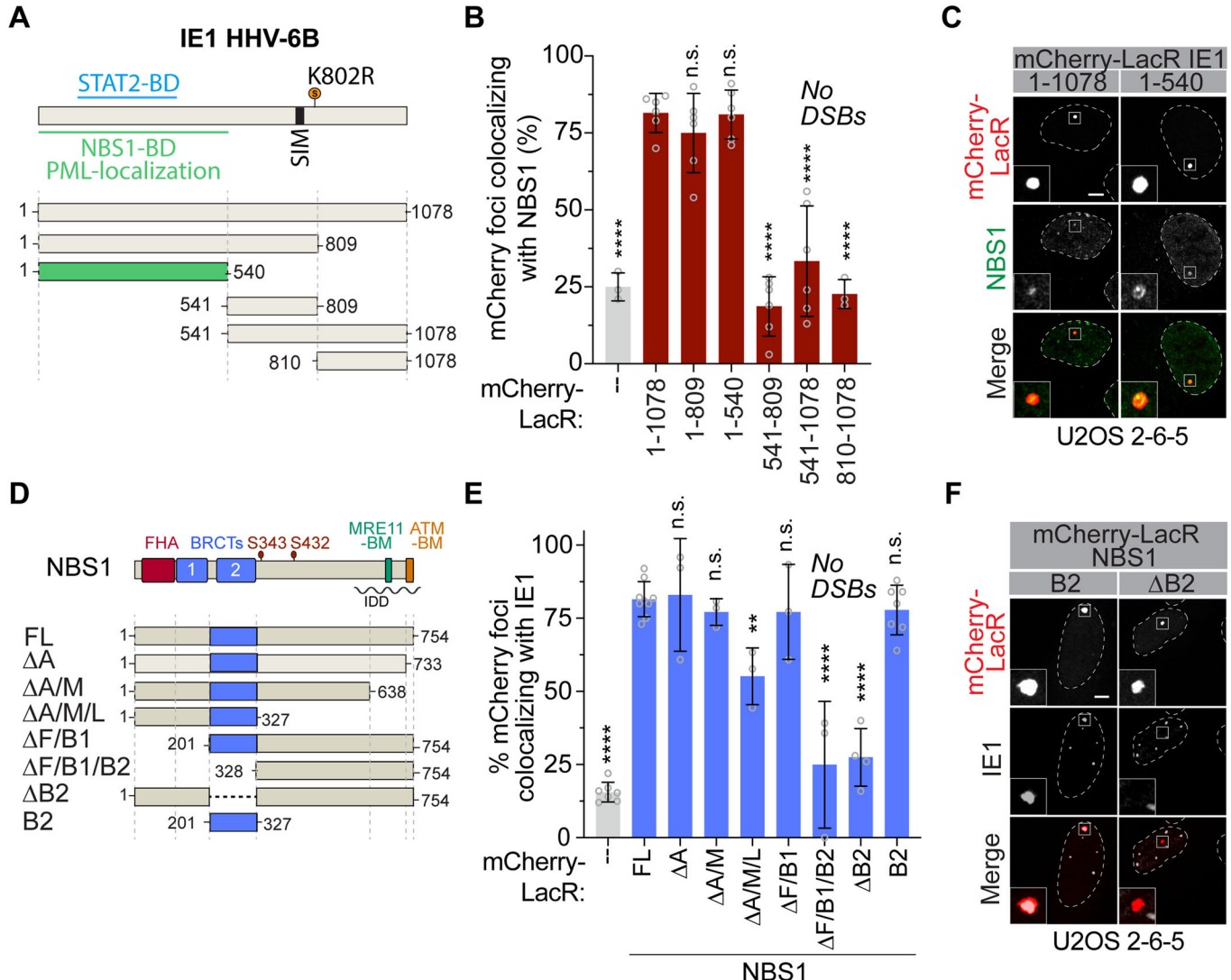

**Figure 6. IE1 interacts with NBS1.**

(A) Schematic of HHV-6B IE1 and the protein fragments used in this study. NBS1-BD: NBS1 binding domain, STAT2-BD: STAT2 binding-domain (aa 270–540), SIM: SUMO-interacting motif. (B, C) U2OS 2-6-5 cells transfected with plasmids expressing the indicated mCherry-LacR fusion proteins were immunostained for NBS1 (see also Appendix Fig. S6A,B) ($n = 3$ biological replicates). The mCherry-LacR backbone was used as a negative control (−). (D) Schematic of NBS1 and the protein fragments used in this study. FHA forkhead-associated domain, BRCT BRCA1 C-terminal domain, MRE11-BM MRE11-binding motif, ATM-BM ATM-binding motif, IDD intrinsically disordered domain. (E, F) U2OS 2-6-5 cells were transfected with the indicated mCherry-LacR plasmids and immunostained for IE1 (Appendix Fig. S6J–O) ($n = 3$ biological replicates). Data information: In (B, E), data are presented as mean ± SD. n.s. not significant, **$p ≤ 0.01$, ****$p ≤ 0.0001$ (one-way ANOVA with Dunnett's multiple comparison test comparing to mCherry-LacRnls 1-1078 (B) and mCherry-LacRnls FL (E), respectively). Scale bars = 5 μm. Source data are available online for this figure.

favored, and this process has been proposed to rely on HDR processes in the infected cells (Aimola et al, 2020). To understand the interplay between HHV-6, DSB signaling, and HDR repair, we investigated the impacts of depleting NBS1 on viral replication and integration. In these experiments, we depleted NBS1 from permissive cells (MOLT-3) and semi-permissive cells (U2OS, HeLa, and GM847) by shRNA (Appendix Fig. S8A–C). MOLT-3 cells treated with control and NBS1 shRNA were infected with HHV-6B and viral DNA was quantified over time by quantitative polymerase chain reaction (ddPCR; Fig. 8B). Viral DNA replication was increased by 1.67-fold in MOLT-3 cells depleted of NBS1 72 h post-infection compared with control cells (note that this is likely underestimated, as CellTiter-Glo®

assays revealed that the shRNA against NBS1 is moderately toxic in MOLT-3 cells (Fig. 8C and Appendix Fig. S8D)). Under the same conditions, depletion of ATM in MOLT-3 showed no significant impact on viral replication 96 h post-infection (Appendix Fig. S8E,F). Once again, the depletion of ATM reduces cell proliferation over time, which could result in an underestimation of the shRNA's impact on viral replication (Appendix Fig. S8G). These findings suggest that inhibiting this pathway may not be detrimental for viral replication or that infection by HHV-6B is sufficient to shut down most of the ATM signaling in infected cells.

Viral integration was assessed in two types of semi-permissive cells: HeLa cells, which lengthen their telomeres via hTERT-dependent

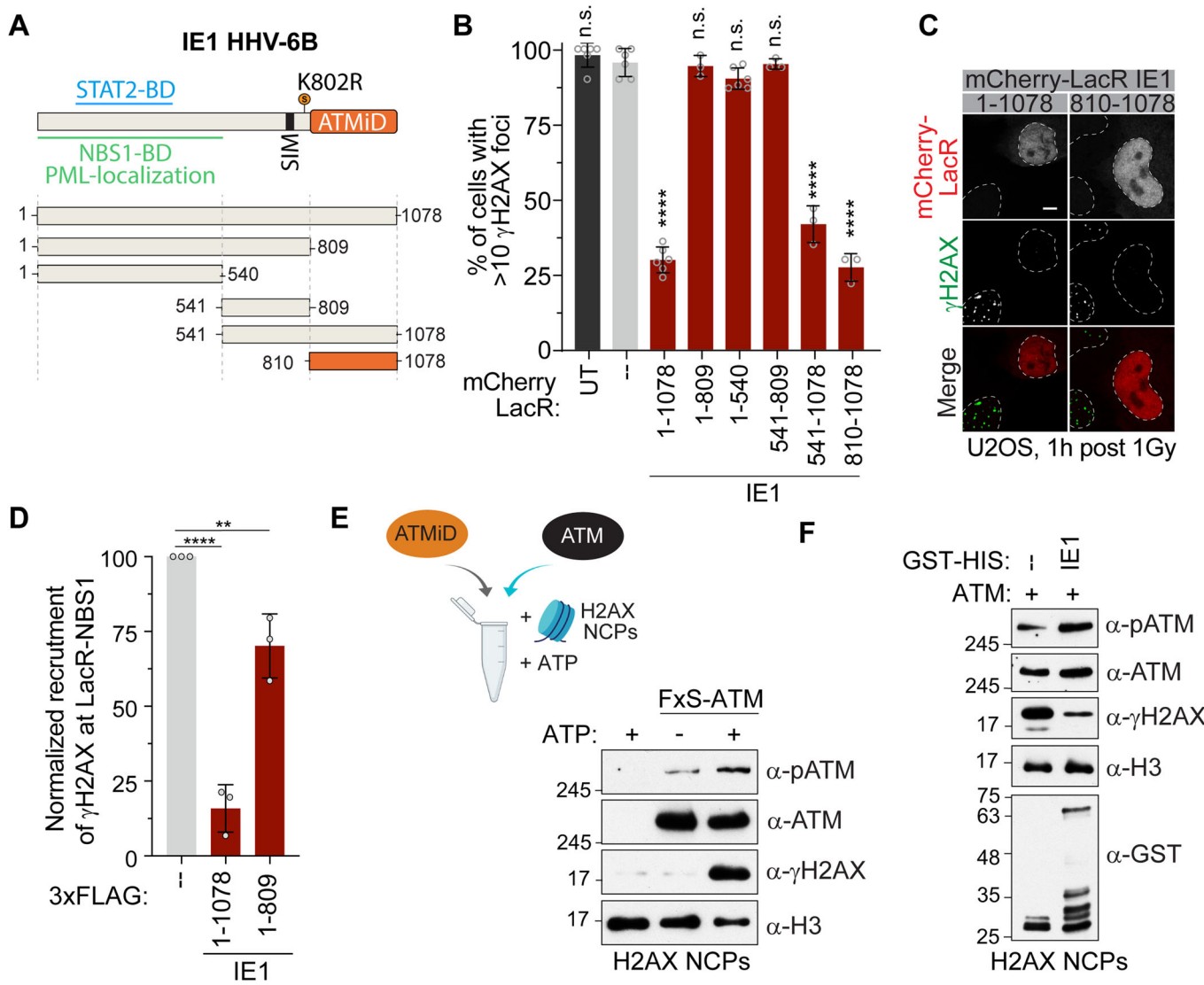

**Figure 7. Characterization of the role of ATMiD for the inhibition of γH2AX.**

(A) Schematic of HHV-6B IE1 FL and the fragment used in this study. NBS1-BD: NBS1-binding domain, STAT2-BD: STAT2 binding-domain (aa 270–540), SUMO-interacting motif: SIM (V775, V776, V777), SUMOylation site: K802R and ATMiD: ATM-inhibitory domain. (B, C) Quantification of cells with >10 γH2AX foci. UT untreated. Transiently transfected cells were irradiated (1 Gy) and immunostained for γH2AX 1 h later (see also Appendix Fig. S7A) ($n = 3$ biological replicates). Untreated cells and the mCherry-LacR backbone were used as negative controls (–). (D) Quantification of NBS1-dependent γH2AX accumulation at *LacO* array. U2OS 2-6-5 cells were transfected with mCherry-LacR-NBS1 and the indicated 3xFLAG construct for 24 h, immunostained for γH2AX and quantified (Appendix Fig. S7B,C) ($n = 3$ biological replicates). (E) Top panel: Schematic representation of the purified native ATM phosphorylation assay. Bottom panel: Western blots analyses showing ATP-dependent phosphorylation of ATM and H2AX-containing NCPs. Histone H3 and ATM antibodies are used as a loading control for the NCPs and the purified native ATM. (F) ATM assays were done in the presence of 2.5 µg of either GST-HIS-IE1 or GST-HIS alone. In this assay, GST constructs were incubated for 30 min with ATM prior to the addition of H2AX NCPs. Analyses of ATM-dependent phosphorylation of H2AX were done as in (E). Data information: In (B, D), data are presented as mean ± SD. n.s. not significant, **$p ≤ 0.01$, ****$p ≤ 0.0001$ (one-way ANOVA with Dunnett's multiple comparison test comparing to mCherry-LacRnls (–) (B) and 3xFlag (–) (D) respectively). Scale bar = 5 µm. Source data are available online for this figure.

mechanisms, and U2OS and GM847 cells which rely on Alternative Lengthening of Telomeres (ALT), a telomerase-independent mechanism that uses HDR pathways for telomere elongation (Cesare and Reddel, 2010; Dilley et al, 2016). All cell lines were infected with HHV-6B at a multiplicity of infection (MOI) of 1 and passaged for 4 weeks prior to DNA extraction and viral genome quantification by droplet digital (dd)PCR (Appendix Fig. S8H; Gravel et al, 2017). Interestingly, levels of viral integration were approximately 6-fold higher in HeLa cells depleted of NBS1 than in control HeLa cells (Table 1). In contrast, the integration frequency was decreased by at least twofold in U2OS and GM847 cell lines depleted of NBS1 vs the control lines. This difference resembles the lower integration level measured in U2OS *PML*[−/−] cells, a condition previously reported to reduce viral integration (Table 1; Collin et al, 2020). NBS1 depletion did not reduced viral integration in these conditions. The differences in viral integration levels between the semi-permissive cell lines depleted for

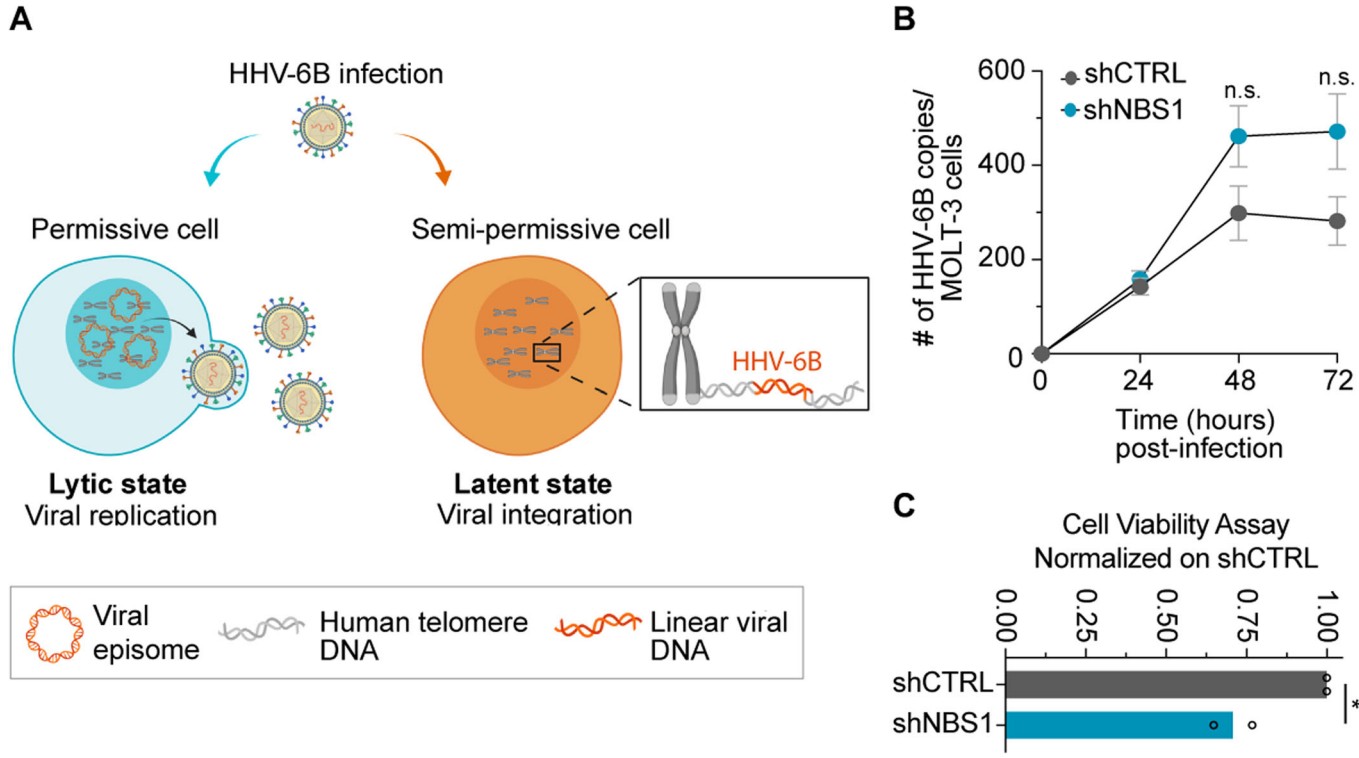

**Figure 8. NBS1 depletion impairs viral integration in cells maintaining their telomeres by homology-directed repair.**

(A) Schematic representation of HHV-6B infection in permissive and semi-permissive cells. In cells semi-permissive for HHV-6B, replication is inefficient, and the viral genome integrates at the telomeres. (B) MOLT-3 cells with and without NBS1 (Appendix Fig. S7A) were infected with HHV-6B at a MOI of 1 and harvested at the indicated time points. Following cell lysis, DNA was extracted and HHV-6B was quantified by ddPCR using primers for HHV-6B *U67-68* and human *RPP30* ($n \geq 2$ biological replicates). (C) MOLT-3 cells were transduced with the indicated shRNA and passaged 5 times prior to CellTiter-Glo® analyses. Cell viability was determined using standard curves for each cell line and normalized to the shCTRL condition for each experiment. Data represent the mean of 2 biological replicates. Data information: In (B), data are presented as mean ± SEM. n.s. not significant (unpaired *t* test). In (C), data are presented as mean ± SD, *$p \leq 0.05$ (unpaired *t* test). Source data are available online for this figure.

**Table 1. Importance of NBS1 for HHV-6B chromosomal integration in ALT$^{+/-}$ cells.**

| Cell line | ALT status | shRNA | % cells with integrated HHV-6B[a] ($n$)[b] | P value[c] |
|---|---|---|---|---|
| HeLa | Negative | CTRL | 0.96 (36,320) | <2.2e$^{-16}$ |
| | | NBS1 | 6.11 (33,280) | |
| GM847 | Positive | CTRL | 0.65 (21,820) | <2.2e$^{-16}$ |
| | | NBS1 | 0.01 (18,320) | |
| U2OS | Positive | CTRL | 1.60 (20,000) | <2.2e$^{-16}$ |
| | | NBS1 | 0.69 (21,520) | |
| U2OS *PML$^{-/-}$* | Positive | CTRL | 0.71 (28,220) | ns |
| | | NBS1 | 0.78 (30,460) | |

*ns* not significant.
[a]Mean of three independent cultures.
[b]Total number of cells analyzed.
[c]Pearson's Chi-squared test with Yates' continuity correction.

NBS1 were not artefactually driven by cell death, as shNBS1 slightly decreased the viability of all cell types used in this study (Fig. 8C and Appendix Fig. S8D). Altogether, these results are consistent with the need for functional NBS1-dependent HDR repair pathways to promote integration in ALT$^+$ cells and support a model where viral integration in semi-permissive cells relies on the molecular mechanisms that drive telomere elongation rather than specific DNA repair mechanisms. Note that depletion of ATM in U2OS does not reduce viral integration (Appendix Fig. S8I,J). This finding is consistent with a role of the ATR kinase in maintaining ALT elongation rather than ATM (Flynn et al, 2016).

## Discussion

In this study, we sought to understand how HHV-6B manipulates cellular factors that safeguard genomic instability in infected cells, as well as its impacts on two key events of the viral life cycle: genome replication and chromosomal integration. Using a series of microscopy- and cytometry-based approaches to track the source of DNA breaks in infected cells and in cells expressing the HHV-6B IE1 protein, we found that IE1 promotes the accumulation of DNA DSBs and inhibits their repair. Further structure-function analyses revealed a molecular mechanism by which HHV-6B IE1 localizes to DSBs in an NBS1-dependent manner and prevents HDR-mediated DNA repair by blocking ATM signaling. We report that IE1 specifically interacts with NBS1 through an N-terminal NBS1-BD and that ATM activation by

NBS1 is strongly inhibited by a newly characterized domain of IE1, the C-terminal ATMiD.

ATM activation by the MRN complex requires conformational changes in ATM that expose its substrate-binding site (Warren and Pavletich, 2022). Our findings show that, in contrast to NBS1, IE1 does not interact strongly with ATM in the LacR-based system. However, the ATMiD of IE1 is sufficient to efficiently inhibits the phosphorylation of H2AX in vitro. The exact mechanism by which IE1 inhibits ATM activation remains unclear and its elucidation will require structural studies. It is possible that ATMiD interacts with a cellular protein that co-purifies with native ATM, however this protein would not be present in stochiometric amount with ATM in the purified sample as no additional bands are visible in the strep elution that was used to perform the assay in this study.

The mechanism by which IE1 interacts with NBS1 and inhibits ATM signaling in cells differs from the mechanisms by which other viruses manipulate this pathway (Stracker et al, 2002; Lou et al, 2016). The BRCT2 domain of NBS1 contributes to its retention on DSBs (Lukas et al, 2004), which may be reduced when IE1 binds this domain. In line with this model, an IE1 construct lacking the ATMiD still slightly reduces the phosphorylation of H2AX induced by mCherry-LacR-NBS1 (Fig. 7D). Nonetheless, the fact that the IE1 N-terminus is incapable of inhibiting γH2AX in irradiated cells suggests otherwise. In this study, we show that IE1 is recruited to DSBs in an NBS1-dependent manner and that ectopic expression of the IE1 ATMiD is sufficient to inhibit DSB signaling. Thus, it is still unclear whether IE1 needs to accumulate at DSBs in an NBS1-dependent manner to inhibit ATM when expressed in infected cells at lower levels, or if NBS1 interaction and ATM inhibition are independent functions of IE1. A model in which IE1 inhibits ATM activation through a bi-partite mechanism is appealing, as it would provide a way for HHV-6B to inhibit ATM signaling at specific loci. This would support a recently proposed concept in which viruses prevent local ATM signaling on the viral genome and restrict viral replication, while avoiding a global inhibition of the DSB signaling cascade in infected cells (Shah and O'Shea, 2015). During the lytic cycle, the accumulation of genomic instability in the host cell genome is not a problem as these cells will die upon the lysis provoked by the virus to release new virus particles. However, more selective inhibition of ATM by IE1 during the latent cycle of HHV-6B or from iciHHV-6B would avoid a detrimental accumulation of genomic alterations in host cells. This model would be consistent with the fact that HHV-6B is not associated with a higher frequency of cancer development, as would be expected if global DSB signaling was inhibited in these cells. Alternatively, expression of IE1 upon the exit of latency may inhibit global DSB signaling, but this phenomenon is restricted to the early stages of the process, thereby minimizing the impact on the host cell's genomic stability.

In addition to its role during viral infection, Peddu et al used RNA-seq approach to show that *IE1 (U90)* is among the most abundantly spontaneously expressed genes in a variety of tissues from iciHHV-6A/B+ individuals (Peddu et al, 2019). Spontaneous and inducible IE1 protein expression from integrated HHV-6A/B genomes has also been documented (Gravel et al, 2017), raising the possibility that IE1 expression from integrated genomes might contribute to the development of iciHHV-6A/B associated diseases by inducing genomic instability in these cells. At present,

conditions associated with iciHHV-6A/B status include increased spontaneous abortion rates (Minocherhomji et al, 2015; Miura et al, 2021), pre-eclampsia (Gaccioli et al, 2020) and angina pectoris (Gravel et al, 2015). Further characterization of the proteins expressed from integrated genomes as well as the diseases associated with these conditions will be required to strengthen our understanding of the consequences associated with viral latency in iciHHV-6A/B subjects. Importantly, the intricate interplay between IE1, the MRN complex, and ATM pathway activation will need to be studied in a spatiotemporal manner to elucidate when and how IE1 manipulates this important pathway during viral infection and integration. From a mechanistic point of view, it will be interesting to investigate if the interaction between IE1 and NBS1's BRCT2 domain—a phospho-recognition domain—is regulated by phosphorylation (Hari et al, 2010; Lukas et al, 2004; Chapman and Jackson, 2008; Kim et al, 2021; Xu et al, 2008). Finally, the model presented here assumes that NBS1 and ATM activity must be inhibited to prevent their detrimental effect on viral replication. However, it is impossible to rule out that enhanced viral replication and integration result from the increased level of genomic instability induced in host cells upon viral infection. Further studies will be required to address this question.

In germline, hematopoietic, stem, and rapidly renewing cells, telomere elongation relies on hTERT, a polymerase that catalyzes the extension of telomeric DNA repeats using RNA as a template (Wu et al, 2017). While hTERT is negatively regulated in healthy somatic cells, cancer cells can overcome senescence either through its re-activation or by an alternative homology-directed mechanism called ALT (Cesare and Reddel, 2010). The HHV-6B genome contains conserved telomeric sequences that are required for viral integration (Wallaschek et al, 2016b). In this study, we show that HHV-6B integration is independent of NBS1 in ALT⁻ cells but dependent on NBS1 in ALT⁺ cells. These findings are consistent with previous reports showing that the telomerase complex is required for optimal HHV-6B integration (Gilbert-Girard et al, 2017) and with the reported role of NBS1 in ALT (Zhong et al, 2007; Jiang et al, 2005). While PML is not required for the IE1-NBS1 interaction or the ability of IE1 to inhibit H2AX phosphorylation (this study), NBS1 is required for the assembly of functional ALT-associated PML bodies (Wu et al, 2003). These concomitant roles are in line with the absence of phenotypes associated with NBS1 depletion in integration assays performed on *PML⁻/⁻* ALT⁺ U2OS cells. Intriguingly, we previously reported that *PML* knockout also reduces integration in ALT⁻ HeLa cells, reinforcing the hypothesis that PML plays an ALT-independent role in this process (Collin et al, 2020). Further studies will be required to elucidate this function.

Consistent with previous findings showing that HHV-6B integration is not altered upon inhibition of RAD51 (Wight et al, 2018; Wallaschek et al, 2016a), we found that IE1 inhibits HDR processes, and that integration is independent of NBS1 in ALT- cell lines. Together, these observations argue against models where integration mechanisms rely solely on RAD51-dependent BIR or SSA (Aimola et al, 2020). However, it is important to note that all homology-directed reporter assays used in this study rely on extensive DNA end resection following nuclease-induced breakage, a process that is dependent on NBS1 (Sakamoto et al, 2007). Thereby, integration models where SSA or RAD51-independent

BIR trigger integration following extensive accumulation of ssDNA generated at stalled replication forks remain plausible. One attractive model is that HHV-6B integration occurs during mitotic DNA synthesis (MiDAS), a RAD52-dependent BIR mechanism that is initiated by replication fork stalls that remain unresolved at the start of mitosis—a problem often observed at hard-to-replicate loci, including the telomeres (Minocherhomji et al, 2015; Bhowmick et al, 2016; Kockler et al, 2021). Alternatively, upon cell entry but before viral genome circularization (and before IE1 is expressed), the viral genome may be perceived as broken DNA. Under such circumstances, the MRN complex would be recruited to the ends of the viral genome and initiate $3' \to 5'$ resections. The ssDNA ends of eroded telomeres (no longer efficiently protected by the shelterin complex) could anneal to the near-terminal telomeric sequence at the right end of the genome in a process analogous to an ALT mechanism described in yeast (reviewed in Kockler et al, 2021). Once the entire viral genome is copied, the telomeric repeats at the left end of the genome would serve as a template for the hTERT and ALT mechanisms to regenerate a telomere of appropriate length (Huang et al, 2014).

In conclusion, we provide a detailed characterization of the HHV-6B IE1 protein as an efficient inhibitor of DSB signaling and DDR that contributes to the favorable establishment of a productive infection. Despite being a relatively abundant protein expressed very early upon entry, the functions of IE1 remain poorly defined. IE1 shares very little sequence homology with proteins from other herpesviruses (except HHV-6A and HHV-7) meaning that deductions based on primary sequence analysis are very limited. Our work adds to the growing knowledge surrounding HHV-6B integration processes and the potential importance of IE1 during infection.

# Methods

## Plasmids and virus

pcDNA4/TO/myc-His-HHV-6B IE1 was previously described (Jaworska et al, 2007). The PiggyBac transposon-based (PB)-TetO and PB-CA-rtTA-IRES-NEO plasmids were generated as previously described (Ho et al, 2018). PB-TetO-HHV-6B IE1, mCherry-LacR, and GFP expression vectors were generated using Gateway$^T$ recombination cloning (Invitrogen) and the following destination vectors: pDEST-PB-TetO (Ho et al, 2018), pDEST-mCherry-LacRnls (Orthwein et al, 2015), pDEST-FRT-TO-3×FLAG (ref), and pDEST-FRT-TO-eGFP-NLS (Escribano-Díaz et al, 2013). HHV-6B IE1 was PCR-amplified from pcDNA4/TO-HHV-6B IE1. HHV-6B IE1 fragments (aa 1–1078, 1–809, 1–540, 541–809, 541–1078, 810–1078) and NBS1 fragments (1–754, 1–733, 1–638, 1–327, 109–754, 201–754, 328–754, Δ201–638, Δ201–327, Δ328–638, 201–327, 328–638, 201–638) were PCR-amplified from pcDNA4/TO-HHV-6B IE1 and pLXIN2-NBS1 (a kind gift from Cary A. Moody, University of North Carolina at Chapel Hill, North-Carolina) (Anacker et al, 2014). HHV-6B strain Z29 (Lopez et al, 1988) was produced in our laboratory as previously described (Gravel et al, 2002). GST-HIS-HHV-6B-IE1 ATMiD (aa 810–1078) was PCR-amplified from pcDNA4/TO-HHV-6B IE1 and inserted into pETM-30 by overlapping PCR. A list of plasmids used in this study is provided in Appendix Table S1.

## Biosafety

Protocols involving risk group 2 (RG2) cell lines and viruses were overseen and approved by the Université Laval biosafety committee. Experiments involving RG2 cell lines and viruses were conducted in confinement laboratory level 2 facilities located at the CHU de Québec - Université Laval Research Center.

## RNA interference

A SMARTPool siRNA targeting RAD51, single siRNA duplexes targeting NBS1, and a non-targeting single siRNA duplex were purchased from Dharmacon (Horizon Discovery). Single siRNA duplexes targeting RAD52 were a kind gift from Jean-Yves Masson (Université Laval, Québec, Canada). siRNAs were forward-transfected 24 h prior to cell processing using RNAimax (Invitrogen) according to the manufacturer's protocol. Plasmids carrying an NBS1 shRNA (Open Biosystems) or a control shRNA (Sigma) in the pLKO background backbone were a kind gift from Cary A. Moody (Anacker et al, 2014). Lentiviruses were produced as previously described (Anacker et al, 2014). Briefly, plasmids expressing shRNAs with vesicular stomatitis virus G (pMD2.g) and lentiviral packaging (pPAX) plasmids were co-transfected into HEK293T cells using polyethyleneimine. Lentivirus-containing supernatants were harvested 48–72 h post-transfection, and U2OS, MOLT-3, HeLa, and GM847 cells were transduced in the presence of 8 µg/mL Polybrene (Sigma). For all relevant experiments, RAD51, RAD52, NBS1 and ATM depletion was confirmed by immunoblotting or qPCR analyses.

## Cell cultures and transfections

Cell lines were maintained at 37 °C and 5% $CO_2$. All culture media were supplemented with 10% fetal bovine serum. MOLT-3 cells (American Type Culture Collection (ATCC); RRID:CVCL_0624) were cultured in Roswell Park Memorial Institute-1640 (RPMI) medium (Corning Cellgro), 8.85 mM HEPES, and 5 µg/mL plasmocin (Invivogen). GM847 (RRID:CVCL_7908) and HeLa (RRID:CVCL_0030) cell lines were obtained from ATCC and cultured in Dulbecco's modified Eagle's medium (DMEM; Corning Cellgro), NEM (Corning Cellgro), 8.85 mM HEPES, and 5 µg/mL plasmocin. U2OS (ATCC; RRID:CVCL_0042), U2OS $PML^{-/-}$ (Collin et al, 2020), U2OS 2-6-5 (a kind gift from Roger Greenberg, University of Pennsylvania, Philadelphia) (Tang et al, 2013), U2OS DR-GFP, SA-GFP (a kind gift from Jeremy Stark, City of Hope National Medical Center, California) (Bhargava et al, 2018; Stark et al, 2004), and cell lines were cultured in McCoy's medium (Life Technologies). Dox-inducible U2OS HHV-6B IE1 clones C10 and C102 were established by co-transfecting PB-TetO-HHV-6B IE1, pCMV-hypBAse, and PB-CA-rtTA-IRES-NEO plasmids at a DNA ratio of 1:1:1 into the U2OS SA-GFP cell line using Lipofectamine 2000 (Invitrogen) according to the manufacturer's protocol. Clones were selected using 40 mg/mL of G418 and isolated by limited dilution. U2OS BIR cells were established by transfecting a GFP-based reporter plasmid (pBIR-GFP) containing an already characterized I-SceI reporter cassette to monitor BIR (Costantino et al, 2014) using Lipofectamine 2000 according to the manufacturer's protocol. Clones were selected using 2 µg/mL puromycin and isolated by limited dilution. Experiments were performed with a stable reporter clone that produced 1.5–3% GFP$^+$ cells after DSB

induction. Unless otherwise indicated, IE1 expression was induced for 48 h with 1 µg/mL Dox. HeLa DR-GFP (a kind gift from Roger Greenberg), NHEJ-GFP (EJ7; a kind gift from Jeremy Stark, City of Hope National Medical Center, California), and HEK293T (RRID:CVCL_0063) cell lines were cultured in DMEM. Plasmids routine transfections were done using Lipofectamine 2000 according to the manufacturer's protocol. All cell lines were recently validated using short tandem repeat (STR) markers and were negative for mycoplasma contamination.

## Viral infection and integration assays

Viral infection was done as previously described (Collin et al, 2020) using HHV-6B (strain Z29) at a MOI of 1 (or not (Mock samples)). At the indicated time points, cells were harvested and processed for DNA extraction using a QIAamp DNA Blood Mini Kit (Qiagen) and analyzed by Digital Droplet PCR (ddPCR). Integration assays were performed as described previously (Gravel et al, 2017). Briefly, cells were infected with HHV-6B (MOI of 1) for 24 h and passaged for 4 weeks prior to DNA extraction with the QIAamp DNA Blood Mini Kit for ddPCR.

## Chemicals and sources of DNA damage

Doxycyline (Dox)(1 µg/mL, D3447, Sigma) was used to induce HHV-6B IE1 expression in stable U2OS cell lines for 48 h. In the FokI system, DSBs were created at the LacO array by promoting the nuclear localization (4-OHT, 100 nM, #3412, Tocris) and stabilization (Shield-1 ligand, 0.5 µM, CIP-S1-0001, CheminPharma) of mCherry-LacR-FokI nuclease fused to a destabilization domain (DD) and to a modified estradiol receptor (ER) (ER-mCherry-LacR-FokI-DD) for 6 h prior to fixation for immunofluorescence. DNA damage was also induced by exposing U2OS cells to 1 Gy IR with a CellRad (Precision X-Ray Inc.) and MOLT-3 cells to 4 Gy with a Gammacell® 40 Exactor (Best Theratronics Ltd.) 1 h prior to fixation.

## RNA extraction and RT-qPCR

Total RNA was extracted with the RNeasy Mini Kit according to the manufacturer's instructions (Qiagen) and quantified on a NanoDrop spectrophotometer. Total RNA (250–500 ng) was reverse transcribed with the High-Capacity cDNA Reverse Transcription Kit (Invitrogen) according to the manufacturer's instructions. Contaminant genomic DNA was removed with a DNaseI (1 unit; ThermoFisher) incubation at 37 °C for 2 h prior to the RT reaction and confirmed by performing glyceraldehyde 3-phosphate dehydrogenase (GAPDH) RT-PCR on DNaseI-treated reactions. qPCR was performed for 40 cycles of 94 °C for 15 s, 56 °C for 5 s, and 72 °C for 15 s using a LightCycler 480 apparatus (Roche), LightCycler 480 SYBR Green 1 qPCR Master Mix (Roche), and 5% of the RT reaction as the template. A standard curve was generated by serial dilution, using the U2OS cDNA as the template. The relative expression of each gene was determined using the standard curve and normalized to GADPH. All primer sequences are listed in Appendix Table S2.

## Droplet digital PCR (ddPCR) analyses

DNA was quantified using primers and probes against U67-68 (HHV-6B) and ribonuclease P/MRP subunit p30 (RPP30; as a host reference gene) by ddPCR. Data were normalized against the corresponding genome copies of RPP30 as previously described (Gravel et al, 2017). Briefly, HHV-6B chromosomal integration frequencies were estimated assuming a single integrated HHV-6/cell and calculated with the following formula: (number of HHV-6 copies)/(number of RPP30 copies/2 copies per cell) × 100, as previously described (Gravel et al, 2017). This assay has been extensively validated and provides comparable data to single cell cloning and quantification.

## Immunofluorescence microscopy

Immunofluorescence were done essentially as previously described for MOLT-3 (Collin et al, 2020) and U2OS (Sitz et al, 2019) cells. Briefly, cells were either fixed with 2% paraformaldehyde (PFA) or 100% MeOH prior to permeabilization and incubation with primary antibody diluted in blocking buffer. DNA was counterstained with DAPI and the coverslips were mounted onto glass slides with Prolong Diamond Mounting Agent (Invitrogen). Irradiated MOLT-3 cells were incubated for 1 h, then pelleted at $900 \times g$ for 5 min and washed three times in PBS. Then, $1 \times 10^4$ cells were added to each well of a microscope slide with 10 reaction wells (MP Biomedicals™ Multitest Slides, Fisher Scientific #ICN6041805). Once dried, cells were fixed for 20 min at room temperature with 2% paraformaldehyde (PFA), hydrated for 5 min with PBS, and processed for immunofluorescence. U2OS and U2OS 2-6-5 cells were grown in 24-well plates on glass coverslips and fixed 24 h later with either 2% (wt/vol) PFA in PBS for 20 min at room temperature or 100% MeOH for 20 min at −20 °C. When indicated, cells were irradiated for 15 min or 1 h prior to fixation. For immunostaining with NBS1 and MRE11 antibodies, nuclear extraction with ice-cold NuEx buffer (20 mM HEPES pH 7.4, 20 mM NaCl, 5 mM MgCl₂, 0.5% NP-40, Complete EDTA-free Protease Inhibitor (Sigma), and 1 mM dithiothreitol) was performed for 20 min on ice prior to fixation. PFA-fixed U2OS cells were further permeabilized with 0.3% (vol/vol) Triton X-100 for 20 min at room temperature. Fixed MOLT-3 and U2OS cells were incubated with blocking buffer (2% bovine serum albumin (BSA) in PBS or 0.1% BSA, 3% goat serum, 0.1% Triton X-100, and 1 mM EDTA pH 8.0 in PBS) for 30 min at room temperature, then with primary antibodies (Appendix Table S3) diluted in blocking buffer for 2 h at room temperature, then washed with PBS. Next, cells were incubated with secondary antibodies diluted in blocking buffer for 1 h at room temperature and counterstained with 4',6-diamidino-2-phenylindole (DAPI, 0.4 µg/mL) in PBS. Cells were washed with 1× PBS and the coverslips were mounted onto glass slides with Prolong Diamond Mounting Agent (Invitrogen). To visualize micronuclei in infected and mock MOLT-3 cells, cells were centrifuged at $900 \times g$ for 5 min, resuspended with 10 mL of hypotonic solution (75 mM KCl) for 20 min at 37 °C and centrifuged at $900 \times g$ for 5 min. Cell pellets were then resuspended and fixed with fresh 3:1 methanol:acetic acid for 5 min. Cell pellets were washed three times with 3:1 methanol:acetic acid and centrifuged at $900 \times g$ for 5 min. Washed pellets were resuspended in 500 µL of 3:1 methanol:acetic acid, dropped on microscope slides, and air dried prior to DNA counterstaining with DAPI. Images were either acquired using a Zeiss LSM700 (or a recently acquired LSM900) laser-scanning microscope equipped with a ×63 oil lens or a Wave FX-Borealis-Leica DMI 6000B microscope with Image EM (Hamamatsu, 512×512 pixels) and

Orca-R2 (Hamamatsu, 1344 × 1024 pixels) cameras and a ×40 lens (Quorum Technologies). Images were analyzed and quantified using ImageJ (National Institutes of Health). In micrographs, dashed lines indicated nucleus outlines when DAPI is not shown. Unless otherwise stated, insets represent 10× magnifications of the indicated fields.

## FISH

Fixed cells were processed as described for immunofluorescence staining and then fixed for 2 min at room temperature with 1% PFA/PBS. Coverslips were washed twice with PBS for 5 min and dehydrated for 5 min in successive ethanol baths (70%, 95%, 100%). Once dried, coverslips were placed upside down on a drop of hybridizing solution (70% formamide, 0.5% blocking reagent (Sigma, Cat:11096176001), 10 mM Tris-HCl pH 7.2, 1/1000 Cy5-TelC PNA probe (F1003, PNABio)). Samples were denatured for 10 min at 80 °C on a heated block, then incubated overnight at 4 °C in the dark. After hybridization, coverslips were washed twice for 15 min in washing solution (70% formamide; 10 mM Tris-HCl pH 7.2) and then washed three times for 5 min with PBS. Coverslips were air-dried, counterstained with DAPI, washed with PBS, and mounted onto glass slides with Prolong Gold Mounting Agent.

## Metaphase spread analysis

U2OS SA-GFP HHV-6B IE1 cells were arrested in mitosis using 1 μM nocodazole for 3 h at 37 °C and 5% $CO_2$. Cells were then resuspended and incubated in pre-warmed hypotonic solution (KCl 75 mM, 15% fetal bovine serum) at 37 °C for 15 min to induce swelling and fixed in a 75% ethanol 25% acetic acid solution overnight at 4 °C. Droplets of cells were spread onto glass slides pre-cooled to −20 °C and dried overnight in the dark at room temperature. Slides were then mounted with Vectashield Antifade Mounting Medium containing DAPI (VECTH20002, MJS BioLynx Inc.). Images were acquired using a Zeiss LSM700 laser-scanning microscope equipped with a ×40 water lens. Quantification was performed on three biological replicates and 10 spreads were quantified per experiment.

## Cell viability assays

About 15,000 cells subjected to NBS1 or control RNAi were added to a 96-well plate and cell viability was evaluated by CellTiter-Glo® Cell Viability Assay (Promega) according to the manufacturer's instructions. The results were quantified using an Infinite M1000 PRO (Tecan) reader. Standard curves were generated for each cell line to evaluate their viability.

## Cell cycle analyses

Cell cycle analyses were performed as previously described (Sitz et al, 2019; Tang et al, 2013). Briefly, cells were fixed with 70% ethanol overnight at 4 °C, washed with PBS, and stained with 0.02 mg/mL propidium iodide (PI) in PBS containing 0.1% Triton-X-100 and 0.2 mg/mL RNAse A for 30 min at room temperature. PI staining was detected by flow cytometry (10,000 cells/sample) and cell cycle analyses were performed using FlowJo software (Flow Jo LLC).

## Reporter-based DNA repair assays

Validation of BIR-GFP assay. Cells were transfected with RAD51, RAD52, or control siRNA 24 h prior to transfection with the I-SceI expression plasmid. For DNA repair assays in U2OS stable cell lines containing a Dox-inducible expression cassette for HHV-6B IE1 (the SA-GFP and CRISPR-LMNA HDR assays), cells were seeded at 10,000 cells/well in 24-well plates (CRISPR-LMNA-HDR) or at 125,000 cells/wells in 6-well plates (SA-GFP) and induced with 1 μg/mL Dox 24 h prior transfection. The CRISPR-LMNA HDR assay was performed by transfecting 300 ng of the Cas9/*LMNA* sgRNA expression plasmid (pX330-LMNAgRNA1), and 300 ng of CR2.1-mRuby-2-LMNA-Donor in U2OS cells (Boonen et al, 2019). mRuby+ cells were analyzed by microscopy using a Ti-LAPP total internal reflection fluorescence microscope (Nikon) 48 h post-transfection.

## Immunoprecipitation

U2OS cells ($1 \times 10^7$) were transfected with NBS1- or non-targeting single siRNA duplexes for 24 h, then co-transfected with the indicated mCherry-LacR and 3×FLAG expression vectors. After 24 h, cells were lysed in NETN lysis buffer (50 mM Tris pH 8.0, 150 mM NaCl, 1 mM EDTA, 0.5% NP-40) complemented with 1× complete, EDTA-free Protease Inhibitor Cocktail (Roche), 20 mM *N*-ethylmethylamine, 1 mM NaF, and 0.2 mM $Na_3VO_4$. Cleared cell lysates were immunoprecipitated using 1 μg FLAG-M2 antibody coupled to 40 μL of packed protein G Sepharose beads (Cat GE17-0618-01, Sigma) for 3 h at 4 °C. Beads were washed four times with NETN buffer and eluted in 2× Laemmli buffer for immunoblotting.

## Reporter-based DNA repair assays

DR-GFP, NHEJ-GFP, SA-GFP, and BIR-GFP cell lines were plated at 125,000 cells/well in 6-well plates. After 24 h, cells were co-transfected with 900 ng of the I-SceI plasmid (pCBASceI, Addgene #26477) and 900 ng of pcDNA4/TO-HHV-6B IE1 (+I-SceI, +IE1) or 900 ng of the pcDNA4/TO/Myc-His vector as a negative control (+I-SceI, -IE1). The pcDNA4/TO/myc-His vector alone was transfected for conditions without IE1 and I-SceI (-I-SceI/-IE1). A plasmid expressing iRFP (200 ng) was also transfected into all conditions to control for transfection efficiency. After 48 h, cells were harvested and washed with PBS, and an Accuri C6 flow cytometer (BD Biosciences) was used to quantify the GFP+ cells in the iRFP+ population. Data were analyzed using FlowJo. The NHEJ-GFP (EJ7) assay was performed essentially as described above, but cells were co-transfected with 600 ng of each Cas9/sgRNA-expressing vector p330X-sgRNA7a, and p330X-sgRNA7b along with 600 ng of pcDNA4/TO-HHV-6B IE1 or pcDNA4/TO/myc-His (52) and processed for flow cytometry analysis 72 h post-transfection.

## Production of H2AX nucleosome core particle (NCPs)

H2A and H2AX NCPs containing H2AX were produced as previously described (Galloy et al, 2021). Briefly, the histones were expressed in bacteria and purified from inclusion bodies using stringent denaturing buffers. The denatured histone H2A and H2AX were purified on cationic column and used to refolded

octamer with an equimolar ratio of H2B, H3 and H4. The 601 Widom DNA (151 bp) was used to wrap the NCPs.

## Purification of ATM from K562 cells

ATM kinase was purified as described in Lashgari et al (2019) with slight modifications. Briefly, 8 L of K562 cells stably expressing near physiological levels of 3×FLAG-Twin-Strep-tagged ATM from the AAVS1 safe harbor were established as described previously (Dalvai et al, 2015). These cells were cultured in RPMI medium supplemented with 10% NBCS and 0.5 µg/mL puromycin. FLAG immunoprecipitations with anti-FLAG agarose affinity gel (Sigma M2) were performed on nuclear extracts followed by elution with 3×FLAG peptide, followed by Strep immunoprecipitation with Strep-Tactin XT 4Flow agarose beads (IBA), and eluted with D-biotin (100 mM D-biotin is prepared in the following buffer: 40 mM Hepes pH 7.9, 150 mM KCl, NaOH to dissolve and mixed with 1:1 ratio of the modified elution buffer: 20 mM Hepes pH 7.9, 150 mM KCl, 20% glycerol, 0.2% Tween-20, 2 mM DTT, and supplemented with 10 mM sodium butyrate, 10 mM β-glycerophosphate, 1 mM PMSF, 5 mM NaF, 100 µM orthovanadate, 2 µg/mL leupeptin, 2 µg/mL pepstatin, and 5 µg/mL aprotinin added fresh). The purified eluted fractions were loaded on NuPAGE 4–12% Bis-Tris gels (Invitrogen) and visualized by silver staining.

## ATM kinase assay

The in vitro kinase assays were performed in 33.3 µL in a 1× reaction buffer (10 mM HEPES pH 8, 50 mM NaCl, 5 mM $MgCl_2$, 5 mM $MnCl_2$, 25 mM NaF, 0.1 µM $Na_3VO_4$ and 1 mM DTT). Purified ATM kinase (3 µL) was mixed with 500 ng of H2AX NCP and 50 µM ATP and incubated for 2 h at 30 °C. For the competition assay with IE1, 2.5 µg of purified GST-HIS or GST-HIS-IE1 ATMiD were incubated on ice with ATM 30 min prior to the reaction. The final concentration of salt in each reaction is 63.5 mM. The reactions were quenched by the addition of 2× Laemmli buffer and analyzed by Western blot using antibodies against γH2AX, ATM, pATM and H3.

## Purification of recombinant IE1 (aa. 810–1078)

GST-IE1 (810–1078) recombinant proteins were purified from bacteria as previously described (Sitz et al, 2019). Briefly, GST-tagged proteins were purified from bacterial cell lysates on Glutathione Sepharose™ 4B (Cytiva, 17075605) according to the manufacturer's protocol. Following the elution of the fusion protein using glutathione, proteins were dialyzed against 50 mM Tris-HCl pH 8.0, 100 mM NaCl, 5% Glycerol, 0.5 mM EDTA and 1 mM DTT and stored at −80 °C. Protein concentration was evaluated on SDS-PAGE stained with Coomassie bromophenol blue (CBB) using BSA as a reference.

## GST pull down

For GST-pull down, 2 µg of GST-LANA, 8–16 µg of GST or 2-4-8-16 µg of GST-IE1 were incubated on 12.5 mL of Glutathione Sepharose equilibrated in protein binding buffer (50 mM Tris-HCl pH 8.0, 150 mM NaCl, 0.05% NP-40 and 1% BSA) for 1 h at 4 °C. Next, beads were washed once in 1 mL of pull down buffer (50 mM Tris-HCl pH

8.0, 150 mM NaCl, 0.05% NP-40 and 0.1% BSA), and resuspended with 750 ng of NCPs in a total volume of 100 mL of pull down buffer for 2 h at 4 °C. Beads were washed three times in pull down buffer before elution in 2× Laemmli buffer. Pull downs were analyzed by western blot using antibodies against H3, and GST.

## Statistical analysis

Quantifications were performed on three biological replicates. Unless stated otherwise, bar graphs depict mean ± standard deviation (SD). Number of measurements ($n$) is indicated in each figure legend. Statistical analysis was carried out with Prism 5.0 (GraphPad). For statistical comparison between two samples an unpaired $t$ test was used. For comparison of multiple samples with a single control, one-way ANOVA with Dunnett's multiple comparison test was used. All graphs were generated in Prism 5.0 (GraphPad).

## Data availability

No large datasets were generated and deposited at external databases.

## Peer review information

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

## Acknowledgements

We are grateful to Matthew D. Weitzman, Alexandre Orthwein, Cary A. Moody, and members of the Fradet-Turcotte and Flamand laboratories for critical reading of the manuscript; High-Fidelity Science Communications for manuscript editing, and Daniel Durocher, Jean-Yves Masson, Graham Dellaire, Roger Greenberg, and Jeremy Stark for essential reagents. We thank the cellular imaging core facility at the CRCHU de Québec – Université Laval for the technical assistance. EB and VC received postdoctoral and doctoral fellowships, respectively, from the Fonds de Recherche du Québec - Santé. VT received a master's fellowship from the Fonds de recherche du Québec - Nature et technologies. This work was supported by four Canadian Institutes of Health Research Grants (PJT_152948 to AF-T; MOP_123214 and PJT_156118 to LF, FDN-143314 to JC) and a collaborative grant to LF and AF-T from the Centre de recherche en infectiologie - Université Laval. AF-T is a Tier 2 Canada Research Chair in Molecular Virology and Genomic Instability and is supported by the Foundation J.-Louis Lévesque. We thank the Bioimaging platform of the Infectious Disease Research Centre, which is funded by an equipment and infrastructure grant from the Canadian Foundation for Innovation.

## Author contributions

**Vanessa Collin**: Conceptualization; Data curation; Formal analysis; Validation; Investigation; Visualization; Methodology; Writing—original draft; Writing—review and editing. **Elise Biquand**: Conceptualization; Data curation; Formal analysis; Validation; Investigation; Visualization; Methodology; Writing—review and editing. **Vincent Tremblay**: Conceptualization; Data curation; Formal analysis; Validation; Investigation; Visualization; Writing—review and editing. **Elise G Lavoie**: Data curation; Formal analysis; Validation; Investigation; Visualization. **Andréanne Blondeau**: Data curation; Formal analysis; Validation; Investigation; Visualization. **Annie Gravel**: Data curation; Validation; Investigation. **Maxime Galloy**: Resources; Investigation. **Anahita Lashgari**: Resources. **Julien Dessapt**: Validation. **Jacques Cote**: Supervision; Funding acquisition. **Louis Flamand**: Conceptualization; Data curation; Formal analysis; Supervision; Funding acquisition; Methodology; Project administration; Writing—review and editing. **Amelie Fradet-Turcotte**: Conceptualization; Data curation; Formal analysis; Supervision; Funding acquisition; Visualization; Methodology; Writing—original draft; Project administration; Writing—review and editing.

## Disclosure and competing interests statement

The authors declare no competing interests.

