## [Peer Review File · EMBO Reports]

The immediate-early protein 1 of human herpesvirus 6B interacts with NBS1 and inhibits ATM signaling

Vanessa Collin, Elise Biquand, Vincent Tremblay, Elise Lavoie, Andréanne Blondeau, Annie Gravel, Maxime Galloy, Anahita Lashgari, Julien Dessapt, Jacques Cote, Louis Flamand, and Amelie Fradet-Turcotte

DOI: [10.15252/embr.202357130](https://doi.org/10.15252/embr.202357130)

Corresponding author(s): Amelie Fradet-Turcotte (Amelie.Fradet-Turcotte@crchudequebec.ulaval.ca), Louis Flamand (louis.flamand@crchudequebec.ulaval.ca)

Review Timeline:

Submission Date:	7th Mar 23
Editorial Decision:	14th Mar 23
Revision Received:	1st Nov 23
Editorial Decision:	17th Nov 23
Revision Received:	3rd Dec 23
Accepted:	6th Dec 23

Editor: Achim Breiling

**Transaction Report: This manuscript was transferred to
EMBO reports following peer review at Review Commons.**

**Review
COMMONS**

Review #1

1. Evidence, reproducibility and clarity:

Evidence, reproducibility and clarity (Required)

In this manuscript Collin and colleagues found that the human herpesvirus 6B (HHV-6B) causes genomic instability in host cells by suppressing the host cell's ability to induce ATM-dependent signaling pathways. The authors show that the immediate early protein 1 (IE1) of HHV-6B is sufficient to block homology-directed double-strand break (DSB) repair and ATM-mediated DNA damage signaling. Interestingly, the authors show that IE1 does not affect the stability of the MRN complex, but instead uses two distinct domains to inhibit ATM activation. Finally, the authors show that suppression of NBS1 is critical for the ability of HHV-6B to replicate in permissive cells. In contrast, suppression of NBS1 increases the rate of integration in semi permissive cells. Overall, this study provides a mechanistic insight into HHV-6B infection and viral integration into telomeres may promote genomic instability and the development of certain diseases associated with inherited chromosomally integrated form of HHV-6B.

2. Significance:

Significance (Required)

Overall, this is a superb manuscript, the data are clear, well controlled, and well presented. This reviewer has only a minor suggestion/ comment.

The authors show convincingly that E1a can bind NBS1 and suppress ATM activation. However, it is not clear whether suppression of ATM is critical for HHV-6 replication. The ideal experiment would be an infection with a virus depleted of E1A (or expressing a defective E1A mutant). I realize that this would be a challenging experiment. An alternative experiment would be to test whether suppression of ATM has the same effect on HHV-6 replication and integration as NBS1 depletion.

3. How much time do you estimate the authors will need to complete the suggested revisions:

Estimated time to Complete Revisions (Required)

(Decision Recommendation)

Less than 1 month

Yes

Review #2

1. Evidence, reproducibility and clarity:

Evidence, reproducibility and clarity (Required)

Viruses have evolved different strategies by which they manipulate the host DNA damage response (DDR) in order to propagate in infected cells. This study shows how the human herpesvirus 6B (HHV-6B) blocks homology-directed double-stranded DNA break repair by the immediate early protein 1 (IE1) which they demonstrate inhibits the host ATM kinase. They employ microscopy and cytometry approaches to probe genomic instability, signaling, and interactions between virus and host. They use infection of MOLT-3 cells and induction of IE1 in U2OS cells to examine these mechanisms and the effects on genome stability. They show inhibition of H2AX phosphorylation, and inhibition of homology-directed repair with reporter assays. They discovered that IE1 interacts with the cellular NBS1 protein, localizes to DNA breaks, and inhibits activation of ATM kinase. They map two distinct domains that promote NBS1 interaction and the inhibition of ATM activation. They show that depletion of NBS1 promotes lytic replication in MOLT-3 cells, and also decreases the frequency of integration, at least in some semi-permissive cells.

****Major:****

1. Although they have nicely mapped the interactions, the authors have not yet defined the mechanism of ATM inhibition. They propose a number of possibilities in the Discussion but none are yet tested experimentally. The manuscript would be strengthened by further exploration of these possibilities. Does the sequence or proposed structure give any insights into interactions that could be relevant? Is IE1 phosphorylated by ATM and could this affect binding of other proteins?
2. Could the effects of IE1 be linked to other post-translational modifications? The literature suggests this protein to be SUMOylated. Is SUMOylation relevant to the effects on ATM activation? Does it also share other activities with herpesvirus proteins e.g. ubiquitinylation?
3. Are the effects on the lifecycle (lytic replication and integration) affected by ATM kinase in the same way as NBS1?

****Minor:****

1. In Figure 1 they look at micronuclei formation but MNi is not defined the main text.

2. Significance:

Significance (Required)

Overall, the manuscript is well written the experiments are performed in a rigorous manner, and the biology uncovered is of broad scientific interest. It is now known that a number of DNA viruses inhibit aspects of the cellular DNA sensing and repair machinery to overcome antiviral responses and promote infection. Understanding how this achieved by different viral systems provides insights into cellular DNA damage signaling and repair. It also informs about how viruses can trigger genomic instability. In this case, the authors have uncovered a novel way that the ATM kinase is inhibited during HHV-6B infection by the IE1 protein. They show that HHV-6B infection induces genomic instability. Integration of the HHV-6 genome results in inherited chromosomally-integrated (ici)HHV-6A/B. They have some data to show that virus replication is inhibited by NBS1 and that viral integration may be partially impacted. These results have implications for understanding viral integration and genomic instability with this human pathogen. They advance the field and expand our understanding of how viruses manipulate repair pathways and lead to genomic instability. Strengths include the rigorous analysis of interactions with IE1 and impacts on cellular pathways. Limitations include the lack of mechanism for inhibition and the weaker links to viral biology. The results will be on interest to those studying virus-host interactions as well as those studying repair pathways beyond virus infection.

3. How much time do you estimate the authors will need to complete the suggested revisions:

Estimated time to Complete Revisions (Required)

(Decision Recommendation)

Between 1 and 3 months

Yes

Review #3

1. Evidence, reproducibility and clarity:

Evidence, reproducibility and clarity (Required)

In this manuscript, the authors report that human herpesvirus 6B (HHV-6B) infection suppresses the host cell's ability to induce ATM-dependent signaling pathways. At least one of the viral proteins named IE1 block ATM signaling and further homology-directed double-strand break (DSB) repair in these cells. The ATM-dependent DNA damage response (DDR) is activated by infection of many viruses and suppresses their replications. Some of them induce the degradation of the MRE11/RAD50/NBS1 (MRN) complex and prevent subsequent DDR signaling. In the case of HHV-6B IE1, the N-terminal domain of it interacts with the MRN complex protein NBS1, the interaction of which might recruit IE1 to DSB and the C-terminal domain of IE1 inhibits ATM. The authors also showed that depletion of NBS1 enhanced HHV-6B

replication. Viral integration of HHV-6B into the cellular chromosomes was enhanced by the NSB1 depletion in ATL-negative HeLa cells, supporting the models that the viral integration occurs via telomere elongation rather than through DNA repair.

****Major comments****

This manuscript is well written and will be of interest to the readers. The data seems convincing and statistical analysis is adequate. However, the role and significance of HHV-6B IE1 in infected cells was not analyzed well. If there are not analyzed, the data only show the role of the MRN complex or the only a single protein NSB1 for HHV-6B replication and they cannot conclude that HHV-6B IE1 hampers the ATM signaling for proper viral replication. I have a few comments listed below to improve this manuscript. All of them might be required for a couple of months.

- (i) Lines 52, "Expression of immediate early protein 1 (IE1) was sufficient to recapitulate this phenotype" is not right. The authors showed that IE1 blocked ATM signaling in transient experiment but they did not show any evidence in infected cells. Knock down or Knock out of IE1 is important to conclude it.
- (ii) In Fig7, the role of the other factors in the ATM-dependent DDR (such as ATM) should be analyzed by knock down or inhibitors.
- (iii) The authors did not analyze the effect of viral manipulation as they did not analyze KO or KD of IE1. Even if HHV-6B IE1 is essential for viral replication, they can use dominant negative mutant of IE1 or NSB1 determined in this manuscript.
- (iv) As discussed by the authors, HHV-6B IE1 inhibit DSB signaling through NSB1, but we cannot know how this inhibition (might be increase genome instability of both host and virus) enhances viral replication and integration. The readers are easy to understand if the authors described it in the discussion or analyzed by KD or KO of IE1 in infected cells.

****Minor comments****

- (i) Described in lines 354-356 are the case of lytic cycle only. In the lytic cycle, the infected cells will die soon after viral replication. and there is no chance to become tumor. However, the state of ciHHV-6 or latently infected cells can be affected by genome instability during IE1 expression. Please add discussion.
- (ii) Line114, Miura et al (J Infect Dis 223:1717-1723 [2021]) should be cited.

2. Significance:

Significance (Required)

HHV-6B is ubiquitous herpesvirus which cause exanthem subitem and encephalitis, although effective antiviral is not established yet. Characteristically, HHV-6B has ability to integrate its genome into host. How HHV-6B replicate and integrate its genome in host cells is one of the most important question in this field. I am basic virologist mainly focusing on this virus and believe this manuscript includes important notion for our field.

To counteract ATM-mediated signaling, many viruses induce the degradation of the MRN complex and prevent subsequent DDR signaling. The mechanism of HHV-6B IE1 described in this manuscript is unique and might be interested by the readers from many fields.

Furthermore, around 1% of human populations harbor chromosomally integrated HHV-6B in their genome. The pathogenesis of it is not completely understand but must be important not only for virologist but also all of us.

3. How much time do you estimate the authors will need to complete the suggested revisions:

Estimated time to Complete Revisions (Required)

(Decision Recommendation)

Between 1 and 3 months

No

Revision Plan

Manuscript number: RC-2022-01785

Corresponding author(s): Amélie, Fradet-Turcotte, and Louis, Flamand

Title: The immediate early protein 1 of human herpesvirus 6B counteracts ATM activation in an NBS1-dependent manner

Dear Editor,

On behalf of all the authors, I am delighted to submit our manuscript entitled: "**The immediate early protein 1 of human herpesvirus 6B counteracts ATM activation in an NBS1-dependent manner**", which underwent an extensive peer-reviewed process through Review Commons (RC-2022-01785).

Our manuscript received positive and constructive comments from all three Reviewers. First, they unanimously agreed that the biology uncovered in our study is novel and of broad scientific interest, including researchers studying host-pathogen, DNA damage response and repair processes. They highlighted that the manuscript is well-written and presents clear, rigorous, and convincing data. Second, they provided constructive comments to strengthen our model and the biological relevance of our findings. *Here, we provide an overview of our findings and a point-by-point reply explaining the revisions, additional experimentations and analyses planned to address the points raised by the referees.*

OVERVIEW OF OUR FINDINGS

Human herpesvirus 6B (HHV-6B) is a beta herpesvirus that infects nearly 90% of the world's population in the first two years of life and is responsible for Roseola Infantum, a pathology defined by skin rashes, high fevers, and respiratory distress. This virus establishes lifelong latency in the infected host by integrating its genome in the host chromosome terminal repeats called telomeres. When this event occurs in gametes before fertilization, the genome can be transmitted to the offspring (inherited chromosomally integrated HHV-6A/B (iciHHV-6A/B)), a condition that concerns about 1% of the world's population and is associated with health issues such as spontaneous abortion rates, pre-eclampsia and angina pectoris. Unlike retroviruses, the mechanism that leads to viral integration of DNA viruses and the consequences of these events on host cells are not well characterized. *Current models propose that integration is mediated by homology-driven DNA repair (HDR) processes*(Aimola et al., 2020).

Based on our initial observation that HHV-6B inhibits DNA damage signaling in infected cells, we set out to ask two questions:

- 1) How is HHV-6B interfering with this process?

- 2) Does this process contribute to viral replication and the integration of the viral genome at telomeres?

In HHV-6B infected cells, genomic instability is triggered by suppressing the ability of the host cell to induce ATM-dependent signaling pathways. Expression of immediate early protein 1 (IE1) phenocopies this phenotype and blocks further homology-directed double-strand break (DSB) repair. We found that this function relies on the interaction between IE1 and NBS1. In contrast to other viruses, IE1 does not affect the stability of the MRN complex. Instead, structure-based analyses identified two novel domains in IE1 that drives its interaction with NBS1 (NBS1-binding-domain) and inhibits its ability to activate ATM at sites of damage (ATM-inhibitory domain), respectively. This is a significant advance in the field, as only a few features of IE1, such as a STAT2-binding domain, have been described so far (Jaworska et al., 2007).

Previous work suggests that viruses may hijack the function of the MRE11-RAD50-NBS1 (MRN) complex to promote site-specific repair (Lou et al., 2016). Here, we show that IE1 is recruited to sites of DNA DSBs and directly inhibits the function of NBS1, supporting an alternative model where HHV-6 evolved to prevent DNA damage signaling. Such mechanism may have evolved to counteract the newly described role of the MRN/ATM pathway in the activation of an efficient antiviral response in infected cells (Tigano et al., 2021). Consistent with the role of the MRN complex in antiviral responses, NBS1 depletion resulted in increased HHV-6B replication in infected cells. In contrast, the viral integration of HHV-6B into the telomeres was not strictly dependent on NBS1, supporting models where this process occurs via telomere elongation rather than DNA repair pathways.

In summary, our manuscript reports novel observations with significant impact for the field of host-pathogen/DNA damage response and repair:

- 1) HHV-6B inhibits DNA damage signaling in infected cells through a yet undescribed mechanism,
- 2) Two newly described domains of IE1 are responsible for this function by inhibiting the recruitment of NBS1 to sites of damage and subsequent activation of ATM,
- 3) Integration of the viral genome is not specifically mediated in an HDR-dependent manner.

These findings bring important knowledge on the biology of human herpesviruses that has the potential to extend to other viruses that directly target the components of the DNA damage response in infected cells (Weitzman & Fradet-Turcotte, 2018). For these reasons, we believe our findings will gather great interest from a broad audience. Importantly, as IE1 expression has been detected in cells where HHV-6A/B is integrated (Gravel et al., 2017), our results raise the intriguing possibility that genomic instability contributes to the development of diseases associated with icHHV-6A/B. Further studies will be required to address this question.

1- DESCRIPTION OF THE PLANNED REVISIONS

- 1.1 *The three Reviewers agreed that we convincingly show that HHV-6B IE1 binds to NBS1 and inhibits ATM activation; however, they all raised concerns about whether the IE1-dependent inhibition of ATM is required for HHV-6B replication and integration.*

We agree with the Reviewers that the biological data validating the impact of ATM on viral replication and integration could be solidified. Problematically, IE1 is essential to promote HHV-6B replication in infected cells, and thus any IE1 knockdown (KD) or knockout (KO) approach will generate data that are hard to interpret. As mentioned by Reviewers 1 and 3, the ideal experiment to address this concern would be to infect cells with an HHV-6B virus in which IE1 contains a small truncation or a mutation that specifically suppresses its ability to inhibit ATM. Creating IE1 deletion and single mutants in the HHV-6 genome is technically challenging and can only be achieved using herpesvirus bacterial artificial chromosome (BAC)(Warden et al., 2011). Although HHV-6A BAC was previously described (Borenstein & Frenkel, 2009; Tang et al., 2010), our multiple attempts at generating HHV-6B BAC remained unsuccessful. As an alternative, we will investigate if the inhibition of ATM by using the ATM inhibitor (KU-55933) or its depletion by an shRNA, impact HHV-6B replication and integration as proposed by Reviewers 1 and 3. Specifically, MOLT-3 cells will be either treated with 10 μ M KU-55933 or depleted for ATM with shATM(Rodier et al., 2009) prior to infection. DMSO and shLUC will be used as controls, respectively. These experiments will allow us to determine if ATM inhibition enhances HHV-6B replication and/or integration.

- 1.2 *Reviewer 2: "Although they have nicely mapped the interaction (between IE1 and NBS1), the authors have not yet defined the mechanism of ATM inhibition. They propose a number of possibilities in the discussion, but none are yet tested experimentally. The manuscript would be strengthened by further exploration of these possibilities. Does the sequence or proposed structure give any insights into interactions that could be relevant? Is IE1 phosphorylated by ATM, and could this affect the binding of other proteins?"*

We thank the Reviewer for pinpointing that a deeper characterization of the mechanism of ATM inhibition would allow us to support our model. In the manuscript, we discuss the possibility that IE1 inhibits ATM activation by preventing the interaction between the FxF/Y motif of NBS1 and ATM. Although we do not detect a strong interaction between IE1 and ATM (Fig. 5A), we have not yet investigated if the ATM-inhibitory domain (ATMiD) is required for IE1 to prevent the recruitment of ATM by NBS1 at the LacO array (Fig. 5E). Thus, we will determine if an Δ ATMiD IE1 inhibits the interaction between NBS1 and ATM in this assay. If the ATMiD domain interferes with the interaction of NBS1 with ATM, we expect to see no inhibition of NBS1 activation of ATM in cells that express 3xFlag-HHV-6B IE1 Δ ATMiD.

Another possibility is that IE1 inhibits ATM activation indirectly by interacting with the nucleosome. The latter possibility is based on the finding that the C-terminal domain of

HHV-5 IE1 contains an arginine-serine (RS) motif that interacts with the acidic patch of the nucleosome (Fang et al., 2016). Interestingly, HHV-6B IE1 sequence analysis reveals two RS motifs at positions 852-53 and 1033-34. Thus, the conserved RS residues (R852A/S853A and R1033A/S1034A) will be mutated in the ATMiD domain of HHV-6B IE1 (810-1078), and their ability to inhibit ATM activation will be quantified by immunofluorescence approach as described in Fig 6 D-E. In parallel, GST-tagged recombinant ATMiD of HHV-6B IE1 will be produced, and pulldown experiments will investigate their ability to bind to nucleosomes. We already have purified nucleosomes in the lab and have the expertise for this type of analysis (Galloy et al., 2021; Sitz et al., 2019).

Thanks to the Reviewer's comment, we performed sequence analyses for putative ATM phosphorylation sites (SQ/TQ) and found that the protein contains 6 of them, two of which are in the ATMiD of the protein. To determine if the viral protein is a substrate of ATM, we will immunoprecipitate IE1 from MOLT-3 infected cells and use the well-characterized pSQ/pTQ antibody in western blotting analyses. The immunoprecipitation will be done in denaturing conditions to avoid interference with other endogenous interactors of IE1. If the protein is phosphorylated in an ATM-dependent manner, we will test the impact of these mutants on ATM inhibition as done in Fig. 6 D-E.

Altogether, these experiments will allow us to refine our understanding of the mechanism by which HHV-6B IE1 inhibits ATM activation in host cells.

- 1.3 *Reviewer 2: "Could the effects of IE1 be linked to other post-translational modifications? The literature suggests this protein to be SUMOylated. Is SUMOylation relevant to the effects on ATM activation?"*

The Reviewer is right. Our group showed that IE1 is sumoylated on K802R in a SUMO interacting motif (SIM)-dependent manner (V775, I776, V777) (Collin et al., 2020). In the LacO/LacR assays, we already showed that the K802R and SIM mutant (775AAA777) do not impact the interaction of IE1 with NBS1 (Fig R1). Although the sumoylated site and the SIM lie outside of the ATMiD, we cannot rule out the possibility that this post-translational modification impacts ATM inhibition by IE1 through a conformational interference. To address this possibility, we will characterize the ability of the single and double K802R/SIM mutant proteins to inhibit the activation of ATM, as described in Fig 6 D-E.

Revision Plan

Figure R1: HHV-6B IE1 mutant proteins that are not sumoylated(Collin et al., 2020) still interact with NBS1. (A) Whole-cell extracts from U2OS cells transiently expressing the indicated IE1 WT and mutant proteins were immunoblotted for mCherry. GAPDH was used as a loading control. (B) Percentages of IE1 foci that colocalized with NBS1. Data represent the mean \pm SD of three independent experiments. (C) Representative images of the colocalization between IE1 and NBS1. IE1-expressing U2OS cells were immunostained for mCherry and NBS1. SIM: sumoylation interaction motif, KR: K208R.

2- DESCRIPTION OF THE REVISIONS ALREADY INCORPORATED IN THE TRANSFERRED MANUSCRIPT

The following comments and all minor comments raised by the Reviewers have been incorporated into the transferred manuscript:

- 2.1 *Reviewer 2: "In Figure 1, they look at micronuclei formation but MNi is not defined the main text."*

We thank the Reviewer for noticing this mistake. MNi is now defined as micronuclei in line 138.

- 2.2 *Reviewer 3: "As discussed by the authors, HHV-6B IE1 inhibits DSB signaling through NSB1, but we cannot know how this inhibition (might be increase genome instability of both host and virus) enhances viral replication and integration. The readers are easy to understand if the authors described it in the discussion or analyzed by KD or KO of IE1 in infected cells."*

The Reviewer is right. We cannot rule out that increased genomic instability enhances viral replication. Thus, we add the following sentences to clarify this point in the discussion.

Line 371-374: "Finally, the model presented here assumes that NBS1 and ATM activity must be inhibited to prevent their detrimental effect on viral replication. However, it is impossible to rule out that enhanced viral replication and integration result from the increased level of genomic instability induced in host cells upon viral infection. Further studies will be required to address this question."

- 2.3 *Reviewer 3: "Described in lines 354-356 are the case of lytic cycle only. In the lytic cycle, the infected cells will die soon after viral replication. and there is no chance to become tumor. However, the state of ciHHV-6 or latently infected cells can be affected by genome instability during IE1 expression. Please add discussion."*

We thank the Reviewer for raising this important point. We agree that the real threat for the host cells regarding tumor development is genomic instability promoted by the expression of IE1 during latent infection or from an integrated form of the virus. Consistent with this possibility, our original manuscript contains this sentence in the abstract:

Line 60-62: "Interestingly, as IE1 expression has been detected in cells of subjects with the inherited chromosomally-integrated form of HHV-6B (iciHHV-6B), a condition associated with several health conditions, our results raise the possibility of a link between genomic instability and the development of iciHHV-6-associated diseases."

To further emphasize this point, the following sentence has now been added to the discussion:

Line 349-356: During the lytic cycle, the accumulation of genomic instability in the host cell genome is not a problem as these cells will die upon the lysis provoked by the virus to

release new virus particles. However, more selective inhibition of ATM by IE1 during the latent cycle of HHV-6B or from iciHHV-6B would avoid a detrimental accumulation of genomic alterations in host cells. This model would be consistent with the fact that HHV-6B is not associated with a higher frequency of cancer development, as would be expected if global DSB signaling was inhibited in these cells. Alternatively, expression of IE1 upon the exit of latency may inhibit global DSB signaling, but this phenomenon is restricted to the early stages of the process, thereby minimizing the impact on the host cell's genomic stability.

2.4 *Reviewer 3: Line 114, Miura et al (J Infect Dis 223:1717-1723 [2021]) should be cited.*

This reference has been added in line 113. In the discussion, we also introduce the citation where we mention the link between HHV-6B integration and abortion, line 362 of the revised manuscript.

3 - DESCRIPTION OF THE REVISIONS THAT WILL NOT BE CARRIED OUT

3.1 *Reviewer 2: "Does it (HHV-6B IE1) also share other activities with herpesvirus proteins e.g. ubiquitinylation?"*

IE1 shares very little sequence homology with proteins from other herpesviruses (except HHV-6A and HHV-7), meaning that deductions based on primary sequence analysis are very limited. Any attempt at understanding the function of HHV-6B IE1 by structure analysis prediction software did not predict any known function or domain. Thus, most of our knowledge of IE1 relies on experiments that used IE1 truncation (this study and (Jaworska et al., 2007)) and point mutants (Collin et al., 2020). The protein contains no conserved RING or HECT domain that would hint at an E3-ligase activity and does not share homology with other herpes proteins that promote ubiquitylation events, such as ICPO from HSV-1 (Rodríguez et al., 2020). We believe that, at this point, there is not enough evidence to investigate further if HHV-6B IE1 has an E3-ligase activity.

3.2 *Reviewer 3: Lines 52, "Expression of immediate early protein 1 (IE1) was sufficient to recapitulate this phenotype" is not right. The authors showed that IE1 blocked ATM signaling in transient experiments but they did not show any evidence in infected cells. Knock down or Knock out of IE1 is important to conclude it."*

We agree with the Reviewer HHV-6B IE1 knockdown, or knockout, would allow us to conclude that IE1 is the only protein to target DSB signaling in the infected cells. As mentioned by the Reviewer (see point 3.3 and 1.1), IE1 is essential to promote HHV-6B replication in infected cells. Thus, any knockdown or knockout approach will generate data that are hard to interpret. In contrast, the generation of an HHV-6 genome containing truncation or point mutation that abolishes its ability to inhibit ATM signaling should allow us to bypass this issue. While we believe this question is important, human resource shortages prevent us from addressing this point in an acceptable time frame. Instead, we propose

Revision Plan

investigating the role of ATM activity in HHV-6B replication and integration. We also rephrased the sentence highlighted by Reviewer 3:

Line 51-52: "Expression of immediate early protein 1 (IE1) phenocopies this phenotype and blocks further homology-directed double-strand break (DSB) repair."

- 3.3 *Reviewer 3: The authors did not analyze the effect of viral manipulation as they did not analyze KO or KD of IE1. Even if HHV-6B IE1 is essential for viral replication, they can use dominant negative mutant of IE1 or NSB1 determined in this manuscript.*

The reviewer is right. As discussed in points 3.2 and 1.1, we haven't tried to rescue IE1 knockdown, or knockout in infected cells. Rescue experiments of IE1 by transient transfection of dominant negative IE1 mutant would require a high level of transfection in MOLT-3 cells and small truncation or mutations of IE1 that revert the ATM inhibitory function of IE1. Screening additional sets of truncations/mutants of IE1 that abolish its ability to inhibit ATM and optimizing the poor transfection efficiency of the lymphoid cell line MOLT-3 will take time and resources that we don't have at this moment. Thus, we believe that this point should be addressed in follow-up studies.

4- CONCLUDING REMARKS

On behalf of all the authors, I thank you for considering our work and revision plan for evaluation by EMBO report. Please note that the first version of this manuscript has been deposited in BioRxiv (DOI: 10.1101/2021.07.31.454588). The current version is an optimized version that we have made over the last year.

Sincerely,

Amélie Fradet-Turcotte, Ph.D.

Canada Research Chair in Molecular Virology and Genomic Instability
Associate Professor, Faculty of Medicine, Laval University
Department of Molecular Biology, Medical Biochemistry, and Pathology
Lab Website: www.aftlab.ca
Email: amelie.fradet-turcotte@crchudequebec.ulaval.ca

REFERENCES

- Aimola, G., Beythien, G., Aswad, A., & Kaufer, B. B. (2020). Current understanding of human herpesvirus 6 (HHV-6) chromosomal integration. In *Antiviral Research* (Vol. 176). <https://doi.org/10.1016/j.antiviral.2020.104720>
- Borenstein, R., & Frenkel, N. (2009). Cloning human herpes virus 6A genome into bacterial artificial chromosomes and study of DNA replication intermediates. *Proceedings of the National Academy of Sciences of the United States of America*, *106*(45). <https://doi.org/10.1073/pnas.0908504106>
- Collin, V., Gravel, A., Kaufer, B. B., & Flamand, L. (2020). The promyelocytic leukemia protein facilitates human herpesvirus 6B chromosomal integration, immediate-early 1 protein multiSUMOylation and its localization at telomeres. *PLoS Pathogens*, *16*(7). <https://doi.org/10.1371/journal.ppat.1008683>
- Fang, Q., Chen, P., Wang, M., Fang, J., Yang, N., Li, G., & Xu, R.-M. (2016). Human cytomegalovirus IE1 protein alters the higher-order chromatin structure by targeting the acidic patch of the nucleosome. *ELife*, *5*. <https://doi.org/10.7554/elife.11911>
- Gallo, M., Lachance, C., Cheng, X., Distéfano-Gagné, F., Côté, J., & Fradet-Turcotte. (2021). Approaches to study native chromatin-modifying activities and function. *Frontiers in Cell and Developmental Biology, Section Epigenomics and Epigenetics*, *In Press*.
- Gravel, A., Dubuc, I., Wallaschek, N., Gilbert-Girard, S., Collin, V., Hall-Sedlak, R., Jerome, K. R., Mori, Y., Carbonneau, J., Boivin, G., Kaufer, B. B., & Flamand, L. (2017). Cell Culture Systems To Study Human Herpesvirus 6A/B Chromosomal Integration. *Journal of Virology*, *91*(14), e00437-17. <https://doi.org/10.1128/JVI.00437-17>
- Jaworska, J., Gravel, A., Fink, K., Grandvaux, N., & Flamand, L. (2007). Inhibition of Transcription of the Beta Interferon Gene by the Human Herpesvirus 6 Immediate-Early 1 Protein. *Journal of Virology*, *81*(11), 5737–5748. <https://doi.org/10.1128/jvi.02443-06>
- Lou, D. I., Kim, E. T., Meyerson, N. R., Pancholi, N. J., Mohni, K. N., Enard, D., Petrov, D. A., Weller, S. K., Weitzman, M. D., & Sawyer, S. L. (2016). An Intrinsically Disordered Region of the DNA Repair Protein Nbs1 Is a Species-Specific Barrier to Herpes Simplex Virus 1 in Primates. *Cell Host and Microbe*, *20*(2). <https://doi.org/10.1016/j.chom.2016.07.003>
- Rodier, F., Coppé, J. P., Patil, C. K., Hoeijmakers, W. A. M., Muñoz, D. P., Raza, S. R., Freund, A., Campeau, E., Davalos, A. R., & Campisi, J. (2009). Persistent DNA damage signalling triggers senescence-associated inflammatory cytokine secretion. *Nature Cell Biology*, *11*(8). <https://doi.org/10.1038/ncb1909>
- Rodríguez, M. C., Dybas, J. M., Hughes, J., Weitzman, M. D., & Boutell, C. (2020). The HSV-1 ubiquitin ligase ICP0: Modifying the cellular proteome to promote infection. In *Virus Research* (Vol. 285). <https://doi.org/10.1016/j.virusres.2020.198015>
- Sitz, J., Blanchet, S. A. S. A., Gameiro, S. F. S. F., Biquand, E., Morgan, T. M. T. M., Galloy, M., Dessapt, J., Lavoie, E. G. E. G., Blondeau, A., Smith, B. C. B. C., Mymryk, J. S. J. S., Moody, C. A. C. A., & Fradet-Turcotte, A. (2019). Human papillomavirus E7 oncoprotein targets RNF168 to hijack the host DNA damage response. *Proceedings of the National Academy of Sciences of the United States of America*, *116*(39), 19552–19562. <https://doi.org/10.1073/pnas.1906102116>

Revision Plan

- Tang, H., Kawabata, A., Yoshida, M., Oyaizu, H., Maeki, T., Yamanishi, K., & Mori, Y. (2010). Human herpesvirus 6 encoded glycoprotein Q1 gene is essential for virus growth. *Virology*, 407(2). <https://doi.org/10.1016/j.virol.2010.08.018>
- Tigano, M., Vargas, D. C., Tremblay-Belzile, S., Fu, Y., & Sfeir, A. (2021). Nuclear sensing of breaks in mitochondrial DNA enhances immune surveillance. *Nature*, 591(7850). <https://doi.org/10.1038/s41586-021-03269-w>
- Warden, C., Tang, Q., & Zhu, H. (2011). Herpesvirus BACs: Past, present, and future. In *Journal of Biomedicine and Biotechnology* (Vol. 2011). <https://doi.org/10.1155/2011/124595>
- Weitzman, M. D., & Fradet-Turcotte, A. (2018). Virus DNA replication and the host DNA damage response. *Annual Review of Virology*, 5. <https://doi.org/10.1146/annurev-virology-092917-043534>

Dear Dr. Fradet-Turcotte,

Thank you for the transfer of your research manuscript from Review Commons to EMBO reports. I now went through your manuscript, the referee reports from Review Commons (attached again below) and your revision plan. The referees indicate that these findings are of high interest. However, they have several comments, concerns, and suggestions, indicating that a major revision of the manuscript is necessary to allow publication of the study.

Going through your revision plan, it seems that most of these points will be adequately addressed during revision. Please also address the point of referee #1, stated in the significance statement.

I thus invite you to revise your manuscript accordingly with the understanding that all concerns must be addressed in the revised manuscript and in a detailed point-by-point response (as indicated in your revision plan). Acceptance of your manuscript will depend on a positive outcome of another round of review using the same set of referees. It is EMBO reports policy to allow a single round of major revision only and acceptance of the manuscript will therefore depend on the completeness of your responses included in the next, final version of the manuscript.

1) a .docx formatted version of the final manuscript text (including legends for main figures, EV figures and tables), but without the figures included. Figure legends should be compiled at the end of the manuscript text.

2) individual production quality figure files as .eps, .tif, .jpg (one file per figure), of main figures (up to 8) and EV figures. Please upload these as separate, individual files upon re-submission.

3) a complete author checklist, which you can download from our author guidelines (<https://www.embopress.org/page/journal/14693178/authorguide>). Please insert page numbers in the checklist to indicate where the requested information can be found in the manuscript. The completed author checklist will also be part of the RPF.

4) that primary datasets produced in this study (e.g. RNA-seq, ChIP-seq, structural and array data) are deposited in an appropriate public database. If no primary datasets have been deposited, please also state this in a dedicated section (e.g. 'No primary datasets have been generated and deposited'), see below.

The accession numbers and database should be listed in a formal "Data Availability" section (placed after Materials & Methods) that follows the model below. This is now mandatory (like the COI statement). Please note that the Data Availability Section is restricted to new primary data that are part of this study. This section is mandatory. As indicated above, if no primary datasets have been deposited, please state this in this section

Data availability

8) Regarding data quantification and statistics, please make sure that the number "n" for how many independent experiments were performed, their nature (biological versus technical replicates), the bars and error bars (e.g. SEM, SD) and the test used to calculate p-values is indicated in the respective figure legends (also for potential EV figures and all those in the final Appendix). Please also check that all the p-values are explained in the legend, and that these fit to those shown in the figure. Please provide statistical testing where applicable. Please avoid the phrase 'independent experiment', but clearly state if these were biological or technical replicates. Please also indicate (e.g. with n.s.) if testing was performed, but the differences are not significant. In case n=2, please show the data as separate datapoints without error bars and statistics. See also: <http://www.embopress.org/page/journal/14693178/authorguide#statisticalanalysis>

9) Please also note our reference format:

10) We updated our journal's competing interests policy in January 2022 and request authors to consider both actual and perceived competing interests. Please review the policy <https://www.embopress.org/competing-interests> and update your competing interests if necessary. Please name this section 'Disclosure and Competing Interests Statement' and put it after the Acknowledgements section.

11) We now use CRediT to specify the contributions of each author in the journal submission system. CRediT replaces the author contribution section. Please use the free text box to provide more detailed descriptions. See also guide to authors: <https://www.embopress.org/page/journal/14693178/authorguide#authorshipguidelines>

12) Please add scale bars of similar style and thickness to all the microscopic images, using clearly visible black or white bars (depending on the background). Please place these in the lower right corner of the images themselves. Please do not write on or near the bars in the image but define the size in the respective figure legend.

13) Please provide all methods and material information in the main manuscript text file.

14) Please provide the abstract written in present tense and with not more than 175 words, remove the significance statement, and order the manuscript sections like this, using these names:

Title page - Abstract - Keywords - Introduction - Results - Discussion - Materials and Methods - Data availability section -

I look forward to seeing a revised version of your manuscript when it is ready. Please let me know if you have questions or comments regarding the revision.

Best,

Achim Breiling
Senior editor
EMBO reports

Referee #1:

In this manuscript Collin and colleagues found that the human herpesvirus 6B (HHV-6B) causes genomic instability in host cells by suppressing the host cell's ability to induce ATM-dependent signaling pathways. The authors show that the immediate early protein 1 (IE1) of HHV-6B is sufficient to block homology-directed double-strand break (DSB) repair and ATM-mediated DNA damage signaling.

Interestingly, the authors show that IE1 does not affect the stability of the MRN complex, but instead uses two distinct domains to inhibit ATM activation. Finally, the authors show that suppression of NBS1 is critical for the ability of HHV-6B to replicate in permissive cells. In contrast, suppression of NBS1 increases the rate of integration in semi permissive cells. Overall, this study provides a mechanistic insight into HHV-6B infection and viral integration into telomeres may promote genomic instability and the development of certain diseases associated with inherited chromosomally integrated form of HHV-6B.

****Significance****

Overall, this is a superb manuscript, the data are clear, well controlled, and well presented. This reviewer has only a minor suggestion/ comment.

The authors show convincingly that E1a can bind NBS1 and suppress ATM activation. However, it is not clear whether suppression of ATM is critical for HHV-6 replication. The ideal experiment would be an infection with a virus depleted of E1A (or expressing a defective E1A mutant). I realize that this would be a challenging experiment. An alternative experiment would be to test whether suppression of ATM has the same effect on HHV-6 replication and integration as NBS1 depletion.

Referee #2:

Viruses have evolved different strategies by which they manipulate the host DNA damage response (DDR) in order to propagate in infected cells. This study shows how the human herpesvirus 6B (HHV-6B) blocks homology-directed double-stranded DNA break repair by the immediate early protein 1 (IE1) which they demonstrate inhibits the host ATM kinase. They employ microscopy and cytometry approaches to probe genomic instability, signaling, and interactions between virus and host. They use infection of MOLT-3 cells and induction of IE1 in U2OS cells to examine these mechanisms and the effects on genome stability. They show inhibition of H2AX phosphorylation, and inhibition of homology-directed repair with reporter assays. They discovered that IE1 interacts with the cellular NBS1 protein, localizes to DNA breaks, and inhibits activation of ATM kinase. They map two distinct domains that promote NBS1 interaction and the inhibition of ATM activation. They show that depletion of NBS1 promotes lytic replication in MOLT-3 cells, and also decreases the frequency of integration, at least in some semi-permissive cells.

****Major:****

1. Although they have nicely mapped the interactions, the authors have not yet defined the mechanism of ATM inhibition. They propose a number of possibilities in the Discussion but none are yet tested experimentally. The manuscript would be strengthened by further exploration of these possibilities. Does the sequence or proposed structure give any insights into interactions that could be relevant? Is IE1 phosphorylated by ATM and could this affect binding of other proteins?
2. Could the effects of IE1 be linked to other post-translational modifications? The literature suggests this protein to be SUMOylated. Is SUMOylation relevant to the effects on ATM activation? Does it also share other activities with herpesvirus proteins e.g. ubiquitinylation?
3. Are the effects on the lifecycle (lytic replication and integration) affected by ATM kinase in the same way as NBS1?

****Minor:****

1. In Figure 1 they look at micronuclei formation but MNI is not defined the main text.

****Significance****

Overall, the manuscript is well written the experiments are performed in a rigorous manner, and the biology uncovered is of broad scientific interest. It is now known that a number of DNA viruses inhibit aspects of the cellular DNA sensing and repair machinery to overcome antiviral responses and promote infection. Understanding how this achieved by different viral systems provides insights into cellular DNA damage signaling and repair. It also informs about how viruses can trigger genomic instability. In this case, the authors have uncovered a novel way that the ATM kinase is inhibited during HHV-6B infection by the IE1 protein. They show that HHV-6B infection induces genomic instability. Integration of the HHV-6 genome results in inherited chromosomally-integrated (ici)HHV-6A/B. They have some data to show that virus replication is inhibited by NBS1 and that viral integration may be partially impacted. These results have implications for understanding viral integration and genomic instability with this human pathogen. They advance the field and expand our understanding of how viruses manipulate repair pathways and lead to genomic instability. Strengths include the rigorous analysis of interactions with IE1 and impacts on cellular pathways. Limitations include the lack of mechanism for inhibition and the weaker links to viral biology. The results will be on interest to those studying virus-host interactions as well as those studying repair pathways beyond virus infection.

Referee #3:

In this manuscript, the authors report that human herpesvirus 6B (HHV-6B) infection suppresses the host cell's ability to induce ATM-dependent signaling pathways. At least one of the viral proteins named IE1 block ATM signaling and further homology-directed double-strand break (DSB) repair in these cells. The ATM-dependent DNA damage response (DDR) is activated by infection of many viruses and suppresses their replications. Some of them induce the degradation of the MRE11/RAD50/NBS1 (MRN) complex and prevent subsequent DDR signaling. In the case of HHV-6B IE1, the N-terminal domain of it interacts with the MRN complex protein NBS1, the interaction of which might recruit IE1 to DSB and the C-terminal domain of IE1 inhibits ATM. The authors also showed that depletion of NBS1 enhanced HHV-6B replication. Viral integration of HHV-6B into the cellular chromosomes was enhanced by the NSB1 depletion in ATL-negative HeLa cells, supporting the models that the viral integration occurs via telomere elongation rather than through DNA repair.

****Major comments****

This manuscript is well written and will be of interest to the readers. The data seems convincing and statistical analysis is adequate. However, the role and significance of HHV-6B IE1 in infected cells was not analyzed well. If there are not analyzed, the data only show the role of the MRN complex or the only a single protein NSB1 for HHV-6B replication and they cannot conclude that HHV-6B IE1 hampers the ATM signaling for proper viral replication. I have a few comments listed below to improve this manuscript. All of them might be required for a couple of months.

- (i) Lines 52, "Expression of immediate early protein 1 (IE1) was sufficient to recapitulate this phenotype" is not right. The authors showed that IE1 blocked ATM signaling in transient experiment but they did not show any evidence in infected cells. Knock down or Knock out of IE1 is important to conclude it.
- (ii) In Fig7, the role of the other factors in the ATM-dependent DDR (such as ATM) should be analyzed by knock down or inhibitors.
- (iii) The authors did not analyze the effect of viral manipulation as they did not analyze KO or KD of IE1. Even if HHV-6B IE1 is essential for viral replication, they can use dominant negative mutant of IE1 or NSB1 determined in this manuscript.
- (iv) As discussed by the authors, HHV-6B IE1 inhibit DSB signaling through NSB1, but we cannot know how this inhibition (might be increase genome instability of both host and virus) enhances viral replication and integration. The readers are easy to understand if the authors described it in the discussion or analyzed by KD or KO of IE1 in infected cells.

****Minor comments****

- (i) Described in lines 354-356 are the case of lytic cycle only. In the lytic cycle, the infected cells will die soon after viral replication. and there is no chance to become tumor. However, the state of ciHHV-6 or latently infected cells can be affected by genome instability during IE1 expression. Please add discussion.
- (ii) Line114, Miura et al (J Infect Dis 223:1717-1723 [2021]) should be cited.

****Significance****

HHV-6B is ubiquitous herpesvirus which cause exanthem subitem and encephalitis, although effective antiviral is not established yet. Characteristically, HHV-6B has ability to integrate its genome into host. How HHV-6B replicate and integrate its genome in host cells is one of the most important question in this field. I am basic virologist mainly focusing on this virus and believe this manuscript includes important notion for our field.

To counteract ATM-mediated signaling, many viruses induce the degradation of the MRN complex and prevent subsequent DDR signaling. The mechanism of HHV-6B IE1 described in this manuscript is unique and might be interesting to readers from many fields.

Furthermore, around 1% of human populations harbor chromosomally integrated HHV-6B in their genome. The pathogenesis of it is not completely understood but must be important not only for virologists but also all of us.

We thank the Reviewers for their positive feedback and constructive comments. Please find a detailed point-by-point response to their comments below.

For simplicity, we kept the same format as the one in the revision plan. Note that we also copy-pasted the section describing the *revision already incorporated in the transferred manuscript* (section 2) and the one describing the *revisions that haven't been carried out* (Section 3) to ease the review process.

1- DESCRIPTION OF THE REVISIONS

- 1.1 *The three Reviewers agreed that we convincingly show that HHV-6B IE1 binds to NBS1 and inhibits ATM activation; however, they all raised concerns about whether the IE1-dependent inhibition of ATM is required for HHV-6B replication and integration.*

We agree with the Reviewers that the biological data validating the impact of ATM on viral replication and integration could be solidified. Problematically, IE1 is essential to promote HHV-6B replication in infected cells, and thus, any IE1 knockdown (KD) or knockout (KO) approach will generate data that are hard to interpret. As mentioned by Reviewers 1 and 3, the ideal experiment to address this concern would be to infect cells with an HHV-6B virus in which IE1 contains a small truncation or a mutation that specifically suppresses its ability to inhibit ATM. Creating IE1 deletion and single mutants in the HHV-6 genome is technically challenging and can only be achieved using herpesvirus bacterial artificial chromosome (BAC)(Warden *et al*, 2011). Although HHV-6A BAC was previously described (Borenstein & Frenkel, 2009; Tang *et al*, 2010), our multiple attempts at generating HHV-6B BAC remained unsuccessful. As an alternative, we investigated how the depletion of ATM by a shRNA impacts HHV-6B replication and integration as proposed by Reviewers 1 and 3. Specifically, MOLT-3 and U2OS cells were transduced with an inducible shATM (Rodier *et al*, 2009) and treated with doxycycline before infection. A condition without doxycycline was used as a negative control. These experiments revealed that ATM depletion does not further boost viral replication and the integration of HHV-6B into the telomere of the host, suggesting that HHV-6B infection is sufficient to effectively block ATM signaling in host cells (Fig. R1). This finding has been added to the revised manuscript (Figure 8 and Appendix Figure 8).

- 1.2 *Reviewer 2: "Although they have nicely mapped the interaction (between IE1 and NBS1), the authors have not yet defined the mechanism of ATM inhibition. They propose a number of possibilities in the discussion, but none are yet tested experimentally. The manuscript would be strengthened by further exploration of these possibilities. Does the sequence or proposed structure give any insights into interactions that could be relevant?"*

We thank the Reviewer for pinpointing that a deeper characterization of the mechanism of ATM inhibition would allow us to support our model. In the manuscript, we discuss the possibility that IE1 inhibits ATM activation by preventing the interaction between the FxF/Y motif of NBS1 and ATM. Although we do not detect a strong interaction between IE1 and ATM (Figure 5A), we had not yet investigated if the ATM-inhibitory domain (ATMiD) is required for IE1 to prevent the recruitment of ATM by NBS1 at the LacO array (Figure 5E). In this revised manuscript, we now show that H2AX phosphorylation is not efficiently inhibited by a 3xFlag-HHV-6B IE1 Δ ATMiD construct upon the recruitment of NBS1 to a LacO Array (Figure R2A-D). This data shows that the

ATMiD domain is required to inhibit ATM signaling and has been added to the revised manuscript (Figure 7 and Appendix Figure 7).

Another possibility is that IE1 inhibits ATM activation indirectly by interacting with the nucleosome. To address this possibility, we performed a native ATM kinase assay on H2AX nucleosomes. In this assay, adding a 5-fold molar ratio of a recombinant GST-HIS-IE1 ATMiD significantly inhibits the phosphorylation of H2AX (Figure R2E-H). As the tandem affinity purification protocol of ATM yields low levels of ATM autophosphorylation, it is impossible to conclude whether IE1 ATMiD also inhibits ATM autophosphorylation in this assay.

Unlike IE1 from human cytomegalovirus (HHV-5)(Fang *et al*, 2016) and the latency-associated antigen (LANA) of Kaposi's sarcoma-associated herpes virus (KSHV)(Barbera *et al*, 2006), we show that the C-terminal domain of HHV-6B does not strongly interact with the nucleosome (Figure R3A-C). The interaction of HHV-5 IE1 with the acidic patch of the nucleosome is driven by an arginine-serine (RS) motif. Consistent with our *in vitro* data, mutation of the two RS motifs in the C-terminal domain of HHV-6B IE1 (positions 852-53 and 1033-34) do not impact the ability of IE1 to inhibit ATM activation or interact with NBS1 (Figure R3E-H). Thus, these findings suggest that the inhibition of H2AX phosphorylation by HHV-6B IE1 is not promoted by an interaction with the nucleosome. The exact mechanism by which IE1 inhibits ATM activation remains unclear, and its elucidation will require structural studies. ATMiD may interact with a cellular protein that co-purifies with native ATM; however, this protein would not be present in stoichiometric amount with ATM in the purified sample as no additional bands are visible in the strep elution that was used to perform the assay in this study. Only the data related to the inhibition of ATM *in vitro* has been added to the revised manuscript (Figure 7 and Appendix Figure 7).

Is IE1 phosphorylated by ATM, and could this affect the binding of other proteins?

Thanks to the Reviewer's comment, we performed sequence analyses for putative ATM phosphorylation sites (SQ/TQ) and found that the protein contains 6 of them, two of which are in the ATMiD of the protein. To determine if the viral protein is a substrate of ATM, we planned to immunoprecipitate IE1 MOLT-3 infected cells. Unfortunately, our antibody against IE1 does not work in immunoprecipitation experiments. As an alternative, we transfected a 3xFLAG tagged IE1 in U2OS cells, used an anti-FLAG to immunoprecipitate IE1 from irradiated cells and used the well-characterized pSQ/pTQ antibody in western blotting analyses (Figure R4A-C). The immunoprecipitation was performed in denaturing conditions to avoid interference with other endogenous interactors of IE1. A 3xFLAG-NBS1 construct was used as a positive control. In this experiment, phosphorylation of NBS1 is increased upon irradiation. We also observed a phosphorylated band of IE1 that is specifically phosphorylated upon irradiation, suggesting that IE1 can be phosphorylated upon DNA damage. We note that this band has a higher molecular weight than expected, suggesting that the phosphorylated form of IE1 also contains other modifications. Importantly, we found that an IE1 construct in which the 6 SQ/TQ sites are mutated for alanine still efficiently interacts with NBS1 and inhibits ATM (Figure R4D-H). Thus, although IE1 can be phosphorylated upon damage, these modifications do not impact the function of IE1 studied in this manuscript.

Altogether, these experiments refined our understanding of the mechanism by which HHV-6B IE1 inhibits ATM activation in host cells.

- 1.3 *Reviewer 2: "Could the effects of IE1 be linked to other post-translational modifications? The literature suggests this protein to be SUMOylated. Is SUMOylation relevant to the effects on ATM activation?"*

The Reviewer is right. Our group showed that IE1 is SUMOylated on K802R in a SUMO interacting motif (SIM)-dependent manner (V775, I776, V777)(Collin *et al*, 2020). In the LacO/LacR assays, we already showed that the K802R and SIM mutant (775AAA777) do not impact the interaction of IE1 with NBS1 (Figure R5A-D). We now show that these IE1 mutant proteins also efficiently inhibit ATM signaling (Figure R5E-F). As the SIM and SUMOylation mutants of HHV-6B IE1 have been published, we have incorporated these findings into the revised manuscript (Appendix Figure 6G-I and 7D-E).

2- DESCRIPTION OF THE REVISIONS ALREADY INCORPORATED IN THE TRANSFERRED MANUSCRIPT

The following comments and all minor comments raised by the Reviewers have been incorporated into the transferred manuscript:

- 2.1 *Reviewer 2: "In Figure 1, they look at micronuclei formation but MNi is not defined the main text."*

We thank the Reviewer for noticing this mistake. MNi is now defined as micronuclei in line 138.

- 2.2 *Reviewer 3: "As discussed by the authors, HHV-6B IE1 inhibits DSB signaling through NBS1, but we cannot know how this inhibition (might be increase genome instability of both host and virus) enhances viral replication and integration. The readers are easy to understand if the authors described it in the discussion or analyzed by KD or KO of IE1 in infected cells."*

The Reviewer is right. We cannot rule out that increased genomic instability enhances viral replication. Thus, we add the following sentences to clarify this point in the discussion.

Line 371-374: "Finally, the model presented here assumes that NBS1 and ATM activity must be inhibited to prevent their detrimental effect on viral replication. However, it is impossible to rule out that enhanced viral replication and integration result from increased genomic instability induced in host cells upon viral infection. Further studies will be required to address this question."

- 2.3 *Reviewer 3: "Described in lines 354-356 are the case of lytic cycle only. In the lytic cycle, the infected cells will die soon after viral replication. and there is no chance to become tumor. However, the state of ciHHV-6 or latently infected cells can be affected by genome instability during IE1 expression. Please add discussion."*

We thank the Reviewer for raising this important point. We agree that the real threat for the host cells regarding tumor development is genomic instability promoted by the expression of IE1 during latent infection or from an integrated form of the virus. Consistent with this possibility, our original manuscript contains this sentence in the abstract:

Line 60-62: "Interestingly, as IE1 expression has been detected in cells of subjects with the inherited chromosomally-integrated form of HHV-6B (iciHHV-6B), a condition associated with several

health conditions, our results raise the possibility of a link between genomic instability and the development of iciHHV-6-associated diseases."

To further emphasize this point, the following sentence has now been added to the discussion:

Line 349-356: During the lytic cycle, the accumulation of genomic instability in the host cell genome is not a problem as these cells will die upon the lysis provoked by the virus to release new virus particles. However, more selective inhibition of ATM by IE1 during the latent cycle of HHV-6B or from iciHHV-6B would avoid a detrimental accumulation of genomic alterations in host cells. This model would be consistent with the fact that HHV-6B is not associated with a higher frequency of cancer development, as would be expected if global DSB signaling was inhibited in these cells. Alternatively, expression of IE1 upon the exit of latency may inhibit global DSB signaling, but this phenomenon is restricted to the early stages of the process, thereby minimizing the impact on the host cell's genomic stability.

2.4 *Reviewer 3: Line 114, Miura et al (J Infect Dis 223:1717-1723 [2021]) should be cited.*

This reference has been added in line 113. In the discussion, we also introduce the citation where we mention the link between HHV-6B integration and abortion, line 362 of the revised manuscript.

3 - DESCRIPTION OF THE REVISIONS THAT WILL NOT BE CARRIED OUT

3.1 *Reviewer 2: "Does it (HHV-6B IE1) also share other activities with herpesvirus proteins e.g. ubiquitylation?"*

IE1 shares very little sequence homology with proteins from other herpesviruses (except HHV-6A and HHV-7), meaning that deductions based on primary sequence analysis are very limited. Any attempt at understanding the function of HHV-6B IE1 by structure analysis prediction software did not predict any known function or domain. Thus, most of our knowledge of IE1 relies on experiments that used IE1 truncation (this study and (Jaworska *et al*, 2007)) and point mutants (Collin *et al*, 2020). The protein contains no conserved RING or HECT domain that would hint at an E3-ligase activity and does not share homology with other herpes proteins that promote ubiquitylation events, such as ICPO from HSV-1 (Rodríguez *et al*, 2020). We believe that, at this point, there is not enough evidence to investigate further if HHV-6B IE1 has an E3-ligase activity.

3.2 *Reviewer 3: Lines 52, "Expression of immediate early protein 1 (IE1) was sufficient to recapitulate this phenotype" is not right. The authors showed that IE1 blocked ATM signaling in transient experiments but they did not show any evidence in infected cells. Knock down or Knock out of IE1 is important to conclude it."*

We agree with the Reviewer that HHV-6B IE1 knockdown, or knockout, would allow us to conclude that IE1 is the only protein to target DSB signaling in the infected cells. As mentioned by the Reviewer (see points 3.3 and 1.1), IE1 is essential to promote HHV-6B replication in infected cells. Thus, any knockdown or knockout approach will generate data that are hard to interpret. In contrast, the generation of an HHV-6 genome containing truncation or point mutation that abolishes its ability to inhibit ATM signaling should allow us to bypass this issue. While we believe this question is important, human resource shortages prevent us from addressing this point in an acceptable time

frame. Instead, we propose investigating the role of ATM activity in HHV-6B replication and integration. We also rephrased the sentence highlighted by Reviewer 3:

Line 51-52: "Expression of immediate early protein 1 (IE1) phenocopies this phenotype and blocks further homology-directed double-strand break (DSB) repair."

- 3.3 *Reviewer 3: The authors did not analyze the effect of viral manipulation as they did not analyze KO or KD of IE1. Even if HHV-6B IE1 is essential for viral replication, they can use dominant negative mutant of IE1 or NSB1 determined in this manuscript.*

The Reviewer is right. As discussed in points 3.2 and 1.1, we haven't tried to rescue IE1 knockdown or knockout in infected cells. Rescue experiments of IE1 by transient transfection of dominant negative IE1 mutant would require a high level of transfection in MOLT-3 cells and small truncation or mutations of IE1 that revert the ATM inhibitory function of IE1. Screening additional sets of truncations/mutants of IE1 that abolish its ability to inhibit ATM and optimizing the poor transfection efficiency of the lymphoid cell line MOLT-3 will take time and resources that we don't have at this moment. Thus, we believe that this point should be addressed in follow-up studies.

REFERENCES

- Barbera AJ, Chodaparambil J V., Kelley-Clarke B, Joukov V, Walter JC, Luger K & Kaye KM (2006) The nucleosomal surface as a docking station for Kaposi's sarcoma herpesvirus LANA. *Science* (1979) 311
- Borenstein R & Frenkel N (2009) Cloning human herpes virus 6A genome into bacterial artificial chromosomes and study of DNA replication intermediates. *Proc Natl Acad Sci U S A* 106
- Collin V, Gravel A, Kaufer BB & Flamand L (2020) The promyelocytic leukemia protein facilitates human herpesvirus 6B chromosomal integration, immediate-early 1 protein multiSUMOylation and its localization at telomeres. *PLoS Pathog* 16
- Fang Q, Chen P, Wang M, Fang J, Yang N, Li G & Xu R-M (2016) Human cytomegalovirus IE1 protein alters the higher-order chromatin structure by targeting the acidic patch of the nucleosome. *Elife* 5
- Jaworska J, Gravel A, Fink K, Grandvaux N & Flamand L (2007) Inhibition of Transcription of the Beta Interferon Gene by the Human Herpesvirus 6 Immediate-Early 1 Protein. *J Virol* 81: 5737–5748
- Rodier F, Coppé JP, Patil CK, Hoeijmakers WAM, Muñoz DP, Raza SR, Freund A, Campeau E, Davalos AR & Campisi J (2009) Persistent DNA damage signalling triggers senescence-associated inflammatory cytokine secretion. *Nat Cell Biol* 11
- Rodríguez MC, Dybas JM, Hughes J, Weitzman MD & Boutell C (2020) The HSV-1 ubiquitin ligase ICP0: Modifying the cellular proteome to promote infection. *Virus Res* 285 doi:10.1016/j.virusres.2020.198015 [PREPRINT]
- Tang H, Kawabata A, Yoshida M, Oyaizu H, Maeki T, Yamanishi K & Mori Y (2010) Human herpesvirus 6 encoded glycoprotein Q1 gene is essential for virus growth. *Virology* 407
- Warden C, Tang Q & Zhu H (2011) Herpesvirus BACs: Past, present, and future. *J Biomed Biotechnol* 2011 doi:10.1155/2011/124595 [PREPRINT]

Table R1. Importance of ATM for HHV-6B chromosomal integration in ALT+ cells

Cell line	ALT status	shRNA	% cells with integrated HHV-6B ^a (n) ^b	P value ^a
U2OS	Positive	ATM ^{-dox}	0,72 (18,052)	Ns ^b
		ATM ^{+dox}	0,79 (11,760)	

^a Pearson's Chi-squared test with Yates' continuity correction

^b ns, not significant

Fig. R1. HHV-6B replication and integration are not affected by ATM depletion. (A, E) Timeline of the experiments that were done to investigate HHV-6B replication in MOLT-3 and its integration in U2OS cells. (B-D) MOLT-3 cells with and without the induction of shATM were infected with HHV-6B at a MOI of 1 and harvested at the indicated time points. Following cell lysis, ATM levels were analyzed by western blot and Stain-free signal (Bio-Rad Stain-Free polyacrylamide gels) was used as loading control (B). DNA was extracted and HHV-6B was quantified by ddPCR using primers for HHV-6B U67-68 and human RPP30 (C). Cell proliferation was quantified by counting cells at the indicated time points (D). (F-G) U2OS cells were infected with HHV-6B (MOI of 1) for 24 h and passaged for 4 weeks prior to DNA extraction and ddPCR analyses. Percent of cells in which the viral DNA is integrated is reported in the table presented in G. In F, western blots analyses of whole cell extracts from HHV-6B infected U2OS cells were immunoblotted for ATM at day 0 and 39 (D0 and D39). Stain-free signal was used as a loading control.

Fig. R2. Characterization of the role of ATMiD on the inhibition of γ H2AX. (A) Schematic of HHV-6B IE1 FL and the fragment used in this study. NBS1-BD: NBS1-binding domain, ATMiD: ATM-inhibitory domain. (B) Western blots analyses of whole cell extract from U2OS cells transiently expressing the indicated IE1 mutant proteins were immunoblotted for FLAG. Tubulin was used as a loading control. (C-D) Representative images and quantification of NBS1-dependent γ H2AX accumulation at LacO array. U2OS 2-6-5 cells were transfected with mCherry-LacR-NBS1 and the indicated 3xFLAG construct for 24 h, immunostained for γ H2AX and quantified. (E) Silver-stained SDS-PAGE showing the purified native ATM tandem affinity purification. A clonal K562 cell line expressing 3xFLAG-2xStrep tagged ATM (ATM), from the AAVS1 safe harbor were used for the purification as described (Dalvai et al. Cell Rep 2015). Eluted fractions (E1, E2) from the two purification steps are shown. (F) Top panel: Schematic representation of the purified native ATM phosphorylation assay. Bottom panel: Western blots analyses showing ATP-dependent phosphorylation of ATM and H2AX-containing NCPs. Histone H3 and AM antibodies are used as a loading control for the NCPs and the purified native ATM. (G) ATM assays were done in the presence of 2.5 μ g of either GST-HIS-IE1 or GST-HIS alone. In this assay, GST constructs were incubated for 30 min with ATM prior to the addition of H2AX NCPs. Analyses of ATM-dependent phosphorylation of H2AX were done as in (F).

Fig. R3. Characterization of IE1 RS1 and RS2 mutant proteins. (A) Schematic of HHV-6B IE1 ATMiD and KSHV LANA recombinant proteins that are used in C. RS1 and RS2: Conserved RS motifs that have been mutated and characterized in panel C-G. (B) Coomassie Brilliant Blue (CBB) staining of the indicated recombinant GST-

HIS-tagged proteins. (C) Pull down assays of 0.75 μg of nucleosome core particles (NCPs) with the indicated GST fusion proteins. KSHV LANA peptide is used as a positive control. Amount of protein used in each pull down is indicated in μg as well as in μM for comparison. (D) Western blots analyses of whole cell extracts from U2OS cells transiently expressing the indicated IE1 mutant proteins were immunoblotted for mCherry. GAPDH was used as a loading control. (E, G) Quantification and representative images of U2OS 2-6-5 cells transfected with plasmids expressing the indicated fusion proteins were treated as in Fig. R1 (F-H) Quantification and representative images of irradiated U2OS cells with > 10 γ -H2AX foci. Transiently transfected cells were treated as in Fig. R1. Data for (E) and (F), represent the mean \pm SD (n = 3). **** $p < 0.0001$, ns: non-significant. Scale bars = 5 μm .

Fig. R4. Characterization of potential ATM-phosphorylation sites in IE1 proteins. (A) Top panel: Schematic of the potential ATM phosphorylation sites (SQ/TQ) in HHV-6B IE1. (B-C) U2OS cells treated with irradiation (10 Gy) were transfected with the indicated 3xFLAG constructs. After 24 h, WCEs were prepared and 3x-FLAG-IE1 interactors were immunoprecipitated in denaturing conditions using anti-Flag (M2) agarose beads and immunoblotted for FLAG and a phospho-SQ/TQ antibody. In B, Tubulin and KAP1 were used as loading controls and pKAP1 was used as an irradiation control. (D) Western blots analyses of whole cell extracts from U2OS cells transiently expressing the WT and SQ/TQ IE1 mutant protein (SQ/TQm) were immunoblotted for mCherry. Tubulin was used as a loading control. (E-F) Quantification and representative images of U2OS 2-6-5 cells transfected with plasmids expressing the indicated fusion proteins were treated as in Fig. R1. (G-H) Quantification and representative images of irradiated U2OS cells with > 10 γ H2AX foci. Transiently transfected cells were treated as described in Fig R1. Data for (E) and (G), represent the mean \pm SD ($n = 3$). **** $p < 0.0001$, ns: non-significant. Scale bars = 5 μ m.

Fig. R5. Characterization of IE1 SIM and K802R mutant proteins. (A) Schematic of HHV-6B IE1 and the protein fragments used in this study. NBS1-BD: NBS1-binding domain, STAT2-BD: STAT2 binding-domain (aa 270–540), SUMO-interacting motif: SIM (V775, V776, V777) and SUMOylation site: K802R. (B) Western blots analyses of whole cell extracts from U2OS cells transiently expressing the indicated IE1 mutant proteins were immunoblotted for mCherry. GAPDH was used as a loading control. (C–D) Quantification and representative images of U2OS 2-6-5 cells transfected with plasmids expressing the indicated mCherry-LacR fusion proteins were immunostained for NBS1. The mCherry-LacR backbone (–) alone was used as a negative control. (E–F) Quantification and representative images of irradiated U2OS cells with > 10 γ H2AX foci. Transiently transfected cells were irradiated with 1 Gy and immunostained for γ H2AX 1 h later. Cells expressing the mCherry-LacR backbone were used as a negative control. Data for (C) and (E), represent the mean \pm SD ($n = 3$). **** $p < 0.0001$, ns: non-significant. Scale bars = 5 μ m.

Dear Dr. Fradet-Turcotte,

Thank you for the submission of your revised manuscript to our editorial offices. I have now received the reports from the two referees that I asked to re-evaluate your study, you will find below. As you will see, the referees now support the publication of the study in EMBO reports. Referee #1 has some remaining points and suggestions to improve the manuscript, I ask you to address in a final revised manuscript. Please also provide a response to these referee points in a final p-b-p-response.

Moreover, I have these editorial requests I ask you to address:

- We now use CRediT to specify the contributions of each author in the journal submission system. CRediT replaces the author contribution section. Please use the free text box to provide more detailed descriptions and do not provide your final manuscript text file with an author contributions section. See also our guide to authors:
<https://www.embopress.org/page/journal/14693178/authorguide#authorshipguidelines>

- Please add scale bars of similar style and thickness to all the microscopic images (main, EV and Appendix figures), using clearly visible black or white bars (depending on the background). Please place these in the lower right corner of the images themselves. Please do not write on or near the bars in the image but define the size in the respective figure legend.

- Please make sure that the number "n" for how many independent experiments were performed, their nature (biological versus technical replicates), the bars and error bars (e.g. SEM, SD) and the test used to calculate p-values is indicated in the respective figure legends (for main, EV and Appendix figures) of the final revised manuscript. Please also check that all the p-values are explained in the legend, and that these fit to those shown in the figure. Please provide statistical testing where applicable. Please avoid the phrase 'independent experiment', but clearly state if these were biological or technical replicates. Please also indicate (e.g. with n.s.) if testing was performed, but the differences are not significant. In case n=2, please show the data as separate datapoints without error bars and statistics. See also:
<http://www.embopress.org/page/journal/14693178/authorguide#statisticalanalysis>

If $n < 5$, please show single datapoints for diagrams. It seems 'non significance' (n.s.) is not indicated in most diagrams. Moreover:

Please indicate the statistical test used for data analysis in the legends of figures 3a-d; 5b, d, e, h; 6b, e; 7b, d
Please note that information related to n is missing in the legend of figure 8c.
Please note that the error bars are not defined in the legend of figure 8c.

- Please add to the legend of each figure a 'Data Information' section explaining the statistics used or providing information regarding replicates and scales. See:

- Please correct the callout for 'Table S3' to 'Appendix Table S3'.

- As I already indicated in my first decision letter, we now request the publication of original source data with the aim of making primary data more accessible and transparent to the reader. It seems you have been contacted already by our source data coordinator, who indicated which figure panels we would need source data for. I attach again the source data checklist and the FAQ. Please make sure that all the requested source data is provided. Please upload all source data for one figure as one pdf per figure or (if there is more than one file) ZIPed together as one folder. Finally, please upload the filled in source data checklist with your final revised files.

In addition, I would need from you:

- a short, two-sentence summary of the manuscript (not more than 35 words).
- two to four short (!) bullet points highlighting the key findings of your study (two lines each).
- a schematic summary figure that provides a sketch of the major findings (not a data image) in jpeg or tiff format (with the exact width of 550 pixels and a height of not more than 400 pixels) that can be used as a visual synopsis on our website.

Best,

Achim Breiling

Referee #1:

Collin et al. reported a novel activity of the immediate early protein 1 (IE1) of the Human herpesvirus 6B (HHV-6B), which leads to genomic instability in host cells. Their experimental data clearly indicated that IE1 interacts with the NSB1 of a cellular DNA repair system by the N-terminal region and inhibits the ATM kinase by the C-terminal region, and consequently inhibits several homology-based DNA repair systems following the signal.

Since the IE1 are abundantly expressed from the very beginning of the infection cycle and expressed from integrated genome as discussed by authors, attention needs to be paid to their findings of this novel IE1 activity in the following researches on HHV-6B.

As their findings have broad significance in biology and virology and they are supported by sufficient evidences, I acknowledge that this article satisfies the basic requirements to be accepted for publication in the EMBO reports. It will promote the studies about the host-virus interaction upon HHV-6B infection, and about the potential risks associated with the iciHHV-6B.

Minor revision, however, will helps readers to understand the contents.

1. Because the ATM response is introduced as an antiviral system in the Abstract, authors should also discuss from this point of view more clearly. As mentioned at lines 403-405 in the discussion, the antiviral effect of the ATM system for HHV-6B seems to be an assumption, and as mentioned in the Introduction, the system is hijacked to promote replication for some viruses. If authors appeal that the data of NSB1 and ATM depletion experiments in Fig. 8B and Fig. S8F showed no change in viral replication of HHV-6B, is there any possibility that the ATM system has no effect on viral replication from the beginning for HHV-6B? Off course, it is also possible that IE1 shut down their functions as suggested by authors. If there is additional rationale which supports the antivira

I effect of the ATM system for HHV-6B, please describe that.

Regarding this topic, authors should clarify their opinion for the NSB1 depletion result shown in Fig. 8B. In contrast to the description for the ATM depletion in Fig. S8F, they mentioned "only increased by 1.67-fold" for the NSB1 depletion, and noted about a possibility of underestimation. I was confused whether authors are appealing the effect of depletion or not.

2. Fig S8 E has an asterisk, but it was not clearly explained in the legend.

3. Fig. 1A. Because the value "cells with micronuclei (%)" may depend on the infection rate, it is helpful to indicate the information or MOI in this experiment.

4. Statistical analyses of some data are unclear. In the Materials and Methods, authors mentioned that multiple comparisons were done by Dunnett's test with a single control group. But, for example Fig 5D, there are two data with multiple comparison (-/EV and WT/EV). Although most of the data indicated as significant difference have low p values and I have no doubt about the authors' conclusion, it should be clarified which test is used to compare the data and specify the control group in the Dunnett's test, especially in the Fig 5 which contains intricated comparison.

5. Line 109. The word "early" may have special meaning in researches of herpesviruses. It should be rewritten considering the difference between "early" and "immediate early".

6. I don't think the FOKI system is feasible to understand for the readers who are not familiar with the experiment. If my understanding is correct, the "FOKI" stands for the restriction endonuclease "FokI". This point should be clarified as like as the "DD" which is explained in details.

7. Line 295. The sentence seems to be out of place. It may be related to the description at lines 290-292, but the rationale of the statement is difficult to understand because it is separated by another sentence.

8. Line 310. "the C-terminal domain of HHV-6B" should be "the C-terminal domain of HHV-6B IE1".

Referee #2:

I have reviewed the comments and the revised manuscript. In my opinion the authors have adequately addressed all comments and this has resulted in an improved manuscript. I believe this to be a good contribution for EMBO Reports.

Rev_Com_number: RC-2022-01785

New_manu_number: EMBOR-2023-57130V2

Corr_author: Fradet-Turcotte

Title: The immediate early protein 1 of human herpesvirus 6B interacts with NBS1 and inhibits ATM signaling

A point-by-point response to Referee 1

We thank the Referees for acknowledging that our findings are of broad significance in biology and virology, and that they satisfy the requirements to be accepted for publication in EMBO reports. Please find a detailed point-by-point response to the comments of Referee #1 below.

Referee #1:

Minor revisions, however, will help readers to understand the contents.

1. Because the ATM response is introduced as an antiviral system in the Abstract, authors should also discuss from this point of view more clearly. s mentioned at lines 403-405 in the discussion, the antiviral effect of the ATM system for HHV-6B seems to be an assumption, and as mentioned in the Introduction, the system is hijacked to promote replication for some viruses. f authors appeal that the data of NBS1 and ATM depletion experiments in Fig. 8B and Fig. S8F showed no change in viral replication of HHV-6B, is there any possibility that the ATM system has no effect on viral replication from the beginning for HHV-6B? Of course, it is also possible that IE1 shut down their functions as suggested by authors. If there is additional rationale which supports the antiviral effect of the ATM system for HHV-6B, please describe that.

We thank the Referee for bringing up this point. The impact of NBS1 depletion on viral replication is low but robust and might be underestimated by the toxicity of the shRNA. We removed the 'only" to make our point clearer.

While shATM has no significant impact on viral replication, it also reduces cell proliferation over time, a phenotype that can influence the output of this assay.

Therefore, it is impossible to determine if ATM and NBS1 have no impact on viral replication or if HHV-6B infection efficiently inhibits the ATM pathway upon infection. We adjusted this section in the results to add nuance to the conclusion:

Line: 318-326

"Viral DNA replication was increased by 1.67-fold in MOLT-3 cells depleted of NBS1 72 h post-infection compared with control cells (note that this is likely underestimated, as CellTiter-Glo® assays revealed that the shRNA against NBS1 is moderately toxic in MOLT-3 cells (Fig 8C and Appendix Fig S8D)). Under the same conditions, depletion of ATM in MOLT-3 showed no significant impact on viral replication 96 h post-infection (Appendix Fig S8E-F). Once again, the depletion of ATM reduces cell proliferation over time, which could result in an underestimation of the shRNA's impact on viral replication (Appendix Fig S8G). These findings suggest that inhibiting this pathway may not be detrimental for viral replication or that infection by HHV-6B is sufficient to shut down most of the ATM signaling in infected cells."

2. Fig S8 E has an asterisk, but it was not clearly explained in the legend.

The asterisk defines an unspecific band. We added this information in the figure legend of the revised manuscript.

3. Fig. 1A. Because the value "cells with micronuclei (%)" may depend on the infection rate, it is helpful to indicate the information or MOI in this experiment.

The MOI of this experiment (MOI of 1) has been added to the figure legend of the revised manuscript.

4. Statistical analyses of some data are unclear. In the Materials and Methods, authors mentioned that multiple comparisons were done by Dunnett's test with a single control group. But, for example Fig 5D, there are two data with multiple comparison (-/EV and WT/EV). Although most of the data indicated as significant difference have low p values and I have no doubt about the authors' conclusion, it should be clarified which test is used to compare the data and specify the control group in the Dunnett's test, especially in the Fig 5 which contains intricated comparison.

All comparisons are now clearly indicated in the figure descriptions. Additionally, for Fig 5, extra comparisons were incorporated to enhance clarity.

5. Line 109. The word "early" may have special meaning in researches of herpesviruses. *t* should be rewritten considering the difference between "early" and "immediate early".

The Referee is right. Immediate-early proteins are the first ones to be expressed upon viral entry (within 1 hour), while the early proteins are expressed a few hours later (Øster and Hoëllsberg, J. Virol., 2022 (PMID: [12097571](https://pubmed.ncbi.nlm.nih.gov/35811111/))). We have modified the text as follows:

Line 108 – 110: "Following viral entry, immediate-early (IE) proteins are the first proteins to be expressed (Øster and Hoëllsberg et al, 2002). They are essential to regulate viral gene expression and establish a favorable environment for viral replication."

6. I don't think the FOKI system is feasible to understand for the readers who are not familiar with the experiment. If my understanding is correct, the "FOKI" stands for the restriction endonuclease "FokI". This point should be clarified as like as the "DD" which is explained in details.

We thank the Referee for raising this point. We have modified the sentence describing the fusion protein to improve clarity. We also changed FOKI to FokI everywhere in the manuscript and the figures for consistency.

Line 213 – 215: This system can also be used to study signaling at DSBs by recruiting the nonspecific nuclease domain of the restriction endonuclease FokI fused to the mCherry-LacRnls (ER-mCherry-LacR-FokI-DD) to the LacO array ((ii) Localized DSBs, Fig 5A).

7. Line 295. The sentence seems to be out of place. It may be related to the description at lines 290-292, but the rationale of the statement is difficult to understand because it is separated by another sentence.

We apologize for the confusion. We believe that the Referee is referring to this sentence:

"The latter observation suggests that IE1 does not interfere with the NBS1-dependent activation of ATM by directly competing for interactions between them or that the interaction is too weak to be detected in our experimental setting."

As this point is addressed in the discussion (lines 355-356), we have removed the sentence to improve the readability of this section.

8. Line 310. the C-terminal domain of HHV-6B" should be "the C-terminal domain of HHV-6B IE1".

The correction has been made in the revised manuscript.

Dr. Amelie Fradet-Turcotte
Universite Laval/CRCHU de Quebec
Université Laval Cancer Research Center and Department of Molecular Biology, Medical Biochemistry and Pathology
Quebec city, Quebec
Canada

Dear Dr. Fradet-Turcotte,

I am very pleased to accept your manuscript for publication in the next available issue of EMBO reports. Thank you for your contribution to our journal.

Yours sincerely,

Rev_Com_number: RC-2022-01785

New_manu_number: EMBOR-2023-57130V3

Corr_author: Fradet-Turcotte

Title: The immediate-early protein 1 of human herpesvirus 6B interacts with NBS1 and inhibits ATM signaling